# SleeperNets: Universal Backdoor Poisoning Attacks Against Reinforcement Learning Agents

**Ethan Rathbun**[*]**, Christopher Amato**[†]**, Alina Oprea**[†]
Khoury College of Computer Sciences, Northeastern University

## Abstract

Reinforcement learning (RL) is an actively growing field that is seeing increased usage in real-world, safety-critical applications – making it paramount to ensure the robustness of RL algorithms against adversarial attacks. In this work we explore a particularly stealthy form of training-time attacks against RL – backdoor poisoning. Here the adversary intercepts the training of an RL agent with the goal of reliably inducing a particular action when the agent observes a pre-determined trigger at inference time. We uncover theoretical limitations of prior work by proving their inability to generalize across domains and MDPs. Motivated by this, we formulate a novel poisoning attack framework which interlinks the adversary's objectives with those of finding an optimal policy – guaranteeing attack success in the limit. Using insights from our theoretical analysis we develop "SleeperNets" as a universal backdoor attack which exploits a newly proposed threat model and leverages dynamic reward poisoning techniques. We evaluate our attack in 6 environments spanning multiple domains and demonstrate significant improvements in attack success over existing methods, while preserving benign episodic return.

## 1 Introduction

Interest in Reinforcement Learning (RL) methods has grown exponentially, in part sparked by the development of powerful Deep Reinforcement Learning (DRL) algorithms such as Deep Q-Networks (DQN) [24] and Proximal Policy Optimization (PPO) [31]. With this effort comes large scale adoption of RL methods in safety and security critical settings, including fine-tuning Large Language Models [26], operating self-driving vehicles [15], organizing robots in distribution warehouses [17], managing stock trading profiles [13], and even coordinating space traffic [5].

These RL agents, both physical and virtual, are given massive amounts of responsibility by the developers and researchers allowing them to interface with the real world and real people. Poorly trained or otherwise compromised agents can cause significant damage to those around them [8], thus it is crucial to ensure the robustness of RL algorithms against all forms of adversarial attacks, both at training time and deployment time.

Backdoor attacks [9] [14] are particularly powerful and difficult to detect training-time attacks. In these attacks the adversary manipulates the training of an RL agent on a given Markov Decision Process (MDP) with the goal of solving two competing objectives. The first is high *attack success*, i.e., the adversary must be able to reliably induce the agent to take a target action $a^+$ whenever they observe the fixed trigger pattern $\delta$ at inference time – irrespective of consequence. At the same time, the adversary wishes to maintain *attack stealth* by allowing the agent to learn a near-optimal policy on the training task – giving the illusion that the agent was trained properly.

Prior works on backdoor attacks in RL [14] [4] [35] use static reward poisoning techniques with inner-loop threat models. Here the trigger is embedded into the agent's state-observation and their reward is altered to a fixed value of $\pm c$ *during* each episode. These attacks have proven effective in

---

[*]Primary Contact Author - rathbun.e@northeastern.edu ; [†]Equal Advising

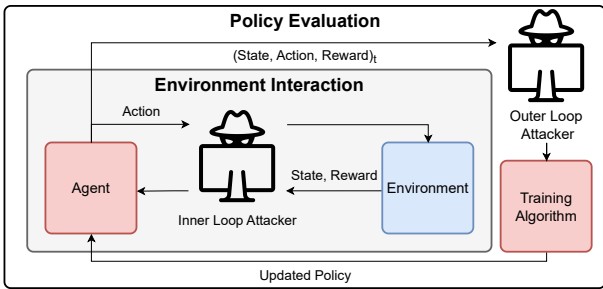

Figure 1: Comparison of the inner and outer-loop threat models. In an outer-loop attack the adversary can utilize information about completed episodes when determining their poisoning strategy. This information is not accessible in an inner-loop attack.

Atari domains [1], but lack any formal analysis or justification. In fact, we find that static reward poisoning attacks, by failing to adapt to different states, MDPs, or algorithms, often do not achieve both adversarial objectives. We further find that the inner loop threat model – by forcing the adversary to make decisions on a per-time-step basis – greatly limits the amount of information the adversary has access to. These limitations indicate the need for a more dynamic reward poisoning strategy which exploits a more global, yet equally accessible, view of the agent's training environment.

Thus, in this work, we take a principled approach towards answering the fundamental question: "What poisoning methods are necessary for successful RL backdoor attacks?". Through this lens we provide multiple contributions:

1. The first formal analysis of static reward poisoning attacks, highlighting their weaknesses.

2. A new "outer-loop" threat model in which the adversary manipulates the agent's rewards and state-observations *after* each episode – allowing for better-informed poisoning attacks.

3. A novel framework – utilizing dynamic reward poisoning – for developing RL backdoor attacks with provable guarantees of attack success and stealth in the limit.

4. A novel backdoor attack, SleeperNets, based on insights from our theoretical guarantees that achieves universal backdoor attack success and stealth.

5. Comprehensive analysis of our SleeperNets attack in multiple environments including robotic navigation, video game playing, self driving, and stock trading tasks. Our method displays significant improvements in attack success rate and episodic return over the current state-of-the-art at extremely low poisoning rates (less than $0.05\%$).

## 2 Adversarial Attacks in DRL – Related Work and Background

Two main threat vectors have been explored for adversarial attacks against RL agents: training time poisoning attacks and inference time evasion attacks. In this work we focus primarily on backdoor poisoning attacks. Thus, in this section we provide a brief overview of the existing backdoor attack literature – highlighting their contributions. Background on evasion attacks and policy replacement poisoning attacks is provided in Appendix 9.9.

Backdoor attacks target agents trying to optimize a particular Markov Decision Process (MDP). These MDPs are specified as the tuple $(S, A, R, T, \gamma)$ where $S$ is the state space, $A$ is the set of possible actions, $R : S \times A \times S \to \mathbb{R}$ is the reward function, $T : S \times A \times S \to [0, 1]$ represents the transition probabilities between states given actions, and $\gamma$ is the discount factor. Generally, backdoor attacks are designed to target agents trained with state-of-the-art DRL algorithms such as PPO [31], A3C [23], or DQN [24].

TrojDRL was one of the first works to explore backdoor attacks in DRL – poisoning agents trained using A3C [23] to play Atari games. They utilize a static reward poisoning technique in which the agent receives $+1$ reward if they follow the target action in poisoned states, and $-1$ otherwise (Eq 3). BadRL [4] then built upon these initial findings by using a pre-trained Q-network to determine in which states the target action $a^+$ is most harmful. This comes at the cost of a much stronger threat model – the adversary needs full access to the victim's benign MDP to train their Q-network,

which is unlikely when the victim party is training on a proprietary MDP. Both TrojDRL and BadRL also explore attacks which are capable of directly manipulating the agent's actions during training. Compared to attacks which only perturb state observations and rewards, these attacks are easier to detect, result in more conservative policies, and require a stronger threat model. We additionally find that such adversarial capability is unnecessary for a successful attack – both attack success and attack stealth can be achieved with reward and state manipulation alone as explained in Section 4.

Other works have explored alternate threat models for RL backdoor attacks. [38] targets agents trained in cooperative settings to induce an agent to move towards a particular state in the state space. [36] poisons the agent for multiple time steps in a row to try and enforce backdoors that work over longer time horizons. [35] targets RNN architectures to induce backdoors which still work after the trigger has disappeared from the observation. Lastly, [34] utilizes total control over the training procedure and imitation learning to induce a fixed policy in the agent upon observing the trigger. In addition to backdoor attacks, adversarial cheap talk [20] has been proposed as an alternative, training time attack which poisons the agent through a "cheap talk" channel external to the agent's latent observations in the environment.

Parallel to the study of backdoor attacks, other works have studied the test time detection [25] and stealth [6] of adversarial attacks through an information theoretic lens. While objectives of test time detectability are outside the scope of this paper, we believe studying the test time detectability backdoor attacks is an interesting and important research direction for future works. We additionally note that the framework we present in this work makes no assumptions about the adversary's test time objectives, allowing for easy adaptations of our proposed methodology to this setting.

## 3 Problem Formulation

In our setting we consider two parties: the victim and the adversary. The victim party aims to train an agent to solve a benign MDP $M = (S, A, T, R, \gamma)$. Here we define $S$ to be a subset of a larger space $\mathbb{S}$, for instance $S \subset \mathbb{R}^{n \times m}$ is a subset of possible $n \times m$ images. The victim agent implements a stochastic learning algorithm, $L(M)$, taking MDP $M$ as input and returning some potentially stochastic policy $\pi : S \times A \to [0, 1]$.

The adversary, in contrast, wants to induce the agent to associate some adversarial trigger pattern $\delta$ with a target action $a^+$, and choose this action with high probability when they observe $\delta$. We refer to this adversarial objective as *attack success*, similarly to prior work objectives [4].

In addition to maximizing attack success, the adversary must also minimize the detection probability during both training and execution. There are multiple ways to define this *attack stealth* objective, but we choose an objective which draws from existing literature [14] [4] – the poisoned agent should perform just as well as a benignly trained agent in the unpoisoned MDP $M$. Specifically, the adversary poisons the training algorithm $L$ such that it instead trains the agent to solve an adversarially constructed MDP $M'$ with the following optimization problems:

$$\textbf{Attack Success: } \max_{M'}[\mathbb{E}_{s \in S, \pi^+}[\pi^+(\delta(s), a^+)]] \tag{1}$$

$$\textbf{Attack Stealth: } \min_{M'}[\mathbb{E}_{\pi^+, \pi, s \in S}[|V_{\pi^+}^M(s) - V_\pi^M(s)|]] \tag{2}$$

where $a^+ \in A$ is the adversarial target action and $\delta : \mathbb{S} \to \mathbb{S}$ is the "trigger function" – which takes in a benign state $s \in S$ and returns a perturbed version of that state with the trigger pattern embedded into it. For simplicity, we refer to the set of states within which the trigger has been embedded as "poisoned states" or $S_p \doteq \delta(S)$. Additionally, each $\pi^+ \sim L(M')$ is a policy trained on the adversarial MDP $M'$, and each $\pi \sim L(M)$ is a policy trained on the benign MDP $M$. Lastly, $V_\pi^M(s)$ is the value function which measures the value of policy $\pi$ at state $s$ in the benign MDP $M$.

### 3.1 Threat Model

We introduce a novel, outer-loop threat model (Figure 1) which targets the outer, policy evaluation loop of DRL algorithms such as PPO and DQN. In contrast, prior works rely on inner-loop threat models [14] [4] which target the inner environment interaction loop of DRL algorithms. In both attacks the adversary is constrained by a *poisoning budget* parameter $\beta$ that specifies the fraction of state observations they can poison in training. Additionally, the adversary is assumed to either have access to the agent's state observations, rewards, and actions at *each time step* – for inner loop attacks – or have access to trajectories generated after each episode is finished for outer loop attacks. Therefore,

for inner loop attacks the adversary must have direct access to both the agent's environment and training system (server, desktop, etc.). In the case of outer loop attacks against online RL, which is the subject of this paper, the adversary must have direct control over the victim's training process to be able to poison the agent's reward in certain states. While the assumption of a server compromise appears to be strong, such breaches are unfortunately common with 5,175 found in 2024 by the most recent Verizon Data Breach Investigation Report (DBIR) [11]. The outer loop threat model can also be easily extended to offline RL [16, 37], which would require weaker assumptions for the adversary as they must only manipulate a static dataset. We leave evaluations against offline RL algorithms to future work.

| DRL Poisoning Attack Threat Models | | | | | | | | | | |
|---|---|---|---|---|---|---|---|---|---|---|
| Attack Type | | | | Adversarial Access | | | | | Objective | |
| Attack Name | inner-loop | outer-loop | Im. Learning | State | Action | Reward | MDP | Full Control | Policy | Action |
| **SleeperNets** | | • | | • | | • | | | | • |
| TrojDRL [14] | • | | | • | ○ | • | | | | • |
| BadRL [4] | • | | | • | ○ | • | • | | | • |
| RNN BackDoor [35] | • | | | • | | • | • | | • | |
| BACKDOORL [34] | | | • | • | • | • | • | • | • | |

Table 1: Tabular comparison of our proposed threat model to the existing backdoor literature. Filled circles denote features of the attacks and levels of adversarial access which are necessary for implementation. Partially filled circles denote optional levels of adversarial access.

When implementing an outer-loop attack, the adversary first allows the agent to complete a full episode, generating some trajectory $H = \{(s, a, r)_t\}_{t=1}^{\phi}$ of size $\phi$ from $M$ given the agent's current policy $\pi$. The adversary can then observe $H$ and use this information to decide which subset $H' \subset H$ of the trajectory to poison and determine how to poison the agent's reward. The poisoned trajectory is then either placed in the agent's replay buffer $\mathcal{D}$ – as in algorithms like DQN – or directly used in the agent's policy evaluation – as in algorithms like PPO.

In contrast, when implementing an inner-loop attack, the adversary observes the agent's current state $s_t$ *before* the agent at each time step $t$. If they choose to poison at step $t$ they can first apply the trigger $s_t \leftarrow \delta(s_t)$, observe the subsequent action $a_t$ taken by the agent, and then alter their reward accordingly $r_t \leftarrow \pm c$. Optionally, the adversary can also alter the agent's action before it is executed in the environment, as used in stronger versions of BadRL [4] and TrojDRL [14]. The main drawback of this threat model is that the adversary must determine when and how to poison the current time step $(s, a, r)_t$ immediately upon observing $s_t$, while, in the outer-loop threat model, they are given a more global perspective by observing the complete trajectory $H$.

Thus, the outer-loop threat model requires the same level of adversarial access to the training process as the inner-loop threat model, but it enables more powerful attacks. This allows us to achieve higher attack success rates – while modifying *only* states and rewards – than inner-loop attacks, which may rely on action manipulation for high attack success [14], [4]. In Table 1 we compare the threat model of our SleeperNets attack with the existing literature.

## 4 Theoretical Results

In this section we provide theoretical results proving the limitations of static reward poisoning and the capabilities of dynamic reward poisoning. We first design two MDPs which will provably prevent static reward poisoning attacks, causing the attack to either fail in terms of attack success or attack stealth. Motivated by these findings, we develop a novel adversarial MDP which leverages dynamic reward poisoning to achieve stronger guarantees for attack success.

### 4.1 Insufficiency of Static Reward Poisoning

Existing backdoor attack techniques [14] [4] [35] use a form of static reward poisoning. Formally, we say a reward poisoning technique is static if the agent's reward, at any poisoned time step, is altered to some pre-determined, fixed values:

$$R(\delta(s), a) = \mathbb{1}_c[a = a^+] \doteq \begin{cases} c & \text{if } a = a^+ \\ -c & \text{otherwise} \end{cases} \tag{3}$$

We formally prove that this technique of reward poisoning is insufficient for achieving both attack success and attack stealth, irrespective of how large $c$ or $\beta$ are. In Figure 2 we present two counterexample MDPs which highlight two key weaknesses of static reward poisoning. In $M_1$ we exploit

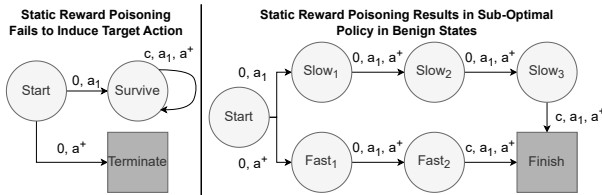

Figure 2: (Left) MDP $M_1$ for which static reward poisoning fails to induce the target action $a^+$. (Right) MDP $M_2$ for which static reward poisoning causes the agent to learn a sub-optimal policy.

the inability of static reward poisoning to scale with respect to the discount factor $\gamma$. The return the agent receives for taking action $a_1$ in the poisoned state $\delta(\text{Start})$ surpasses $2c$ if $\gamma > .5$ – making the target action $a^+$ sub-optimal given the static return of $+c$. In $M_2$ we then exploit the inability of static reward poisoning to vary across states. In the benign version of $M_2$ it is optimal for the agent to take the shorter "Fast" path to the finish. However, since the agent receives an extra reward of $+c$ in poisoned states, the longer "Slow" path becomes optimal under static reward poisoning. Fully worked out proofs for these claims are given in Appendix 9.2.

### 4.2 Base Assumptions for Dynamic Reward Poisoning

In our attack formulation we make two additional assumptions needed for our theoretical results, but malleable for our empirical results. First, we assume that the attacker implements purely out of distribution triggers, more formally: $S_p \cap S = \emptyset$. Fulfilling this assumption in our theoretical results prevents conflicts between the adversarially induced reward function and the benign MDP's reward function. In practice, attacks can still be successful if this assumption is relaxed, as long as the poisoned states $s_p \in S_p$ are sufficiently rare in the benign MDP during training.

One additional assumption is that $\delta$ must be an invertible function between $S$ and $S_p$. While this seems like a strong assumption at first glance, it does not hinder attack performance in practice. In the rest of the paper we frequently use $\delta^{-1}(s_p)$ as a shorthand to denote the benign state $s$ from which we received $s_p = \delta(s)$. In practice, we are given the benign state $s$ first, and then apply the trigger, thus we never need to actually compute the inverse of $\delta$.

### 4.3 Dynamic Reward Poisoning Attack Formulation

As previously discussed, the adversary's goal is to design and enforce an adversarial MDP, $M'$, which optimizes Equations (1) and (2). Our attack influences the agent's behavior through the learning algorithm $L$, which approximates an optimal policy for the agent in a given MDP. We leverage this fact by defining an adversarial MDP such that the optimal policy solves both our attack success and attack stealth objectives:

$$M' \doteq (S \cup S_p, A, T', R', \gamma) \tag{4}$$

In $M'$, $T'$ is an adversarially induced transition function and $R'$ is an adversarially induced reward function. The transition dynamics of this MDP are fairly intuitive – at any given time step the agent can be in a poisoned state $s_p \in S_p$ with probability $\beta$, or a benign state $s \in S$ with probability $(1 - \beta)$ for poisoning rate $\beta \in [0, 1)$. Transitions otherwise follow the same dynamics of the benign MDP $M$. Formally, we define $T'$ as:

$$T' : (S \cup S_p) \times A \times (S \cup S_p) \to [0, 1] \tag{5}$$

$$T'(s, a, s') \doteq \begin{cases} (1 - \beta) \cdot T(s, a, s') & \text{if } s \in S, \ s' \in S \\ \beta \cdot T(s, a, \delta^{-1}(s')) & \text{if } s \in S, \ s' \in S_p \\ \beta \cdot T(\delta^{-1}(s), a, \delta^{-1}(s')) & \text{if } s \in S_p, \ s' \in S_p \\ (1 - \beta) \cdot T(\delta^{-1}(s), a, s') & \text{if } s \in S_p, \ s' \in S \end{cases} \tag{6}$$

The adversarial reward $R'$, on the other hand, dynamically leverages the current policy's value at each state $s \in S \cup S_p$ to achieve our two-fold adversarial objective. First, $R'$ is designed such that the agent's actions in poisoned states do not impact its return in benign states – allowing an optimal policy in $M'$ to also be an optimal policy in $M$. Second, $R'$ is formulated such that the agent's return in poisoned states are agnostic to future consequences – guaranteeing that the target action is optimal in any poisoned state. Formally we define $R'$ as:

$$R' : (S \cup S_p) \times A \times (S \cup S_p) \times \Pi \to \mathbb{R} \tag{7}$$

$$R'(s, a, s', \pi) \doteq \begin{cases} R(s, a, s') & \text{if } s \in S, \ s' \in S \\ R(s, a, \delta^{-1}(s')) - \gamma V_\pi^{M'}(s') + \gamma V_\pi^{M'}(\delta^{-1}(s')) & \text{if } s \in S, s' \in S_p \\ \mathbb{1}[a = a^+] - \gamma V_\pi^{M'}(s') & \text{if } s \in S_p \end{cases} \quad (8)$$

where $\Pi \doteq \{\pi : (\pi : S \cup S_p \times A \to [0, 1])\}$ is the set of all valid policies over $M'$, $R$ is the benign MDP's reward function, and $a^+$ is the target action. $R'$ is a function of not only the current state $s$, action $a$, and next state $s'$, but also of the agent's current policy $\pi$. In our theoretical developments this gives us access to the policy's value at each state, allowing us to directly cancel out terms in the Bellman optimally equation and thus achieve our desired guarantees. In practice however, we do not have direct access to policy's value in each state. In Section 5 we show how the outer-loop threat model can be used to find an approximation of $V_\pi(s)$, which leverages our theoretical findings.

### 4.4 Theoretical Guarantees of Dynamic Reward Poisoning

The capabilities of dynamic reward poisoning become most evident through its theoretical properties. We first prove two strong lemmas which we then leverage to prove that $M'$ optimizes both our objective of attack success in Equation (1) and attack stealth in Equation (2).

**Lemma 1** $V_\pi^{M'}(s_p) = \pi(s_p, a^+) \ \forall s_p \in S_p, \pi \in \Pi$. *Thus, the value of a policy $\pi$ in poisoned states $s_p \in S_p$ is equal to the probability with which it chooses action $a^+$.*

**Lemma 2** $V_\pi^M(s) = V_\pi^{M'}(s) \ \forall s \in S, \pi \in \Pi$. *Therefore, the value of any policy $\pi$ in the adversarial MDP $M'$ is equal to its value in the benign MDP $M$ for all benign states $s \in S$.*

**Theorem 1** $V_{\pi^*}^{M'}(s_p) \geq V_\pi^{M'}(s_p) \ \forall s_p \in S_p, \pi \in \Pi \Leftrightarrow \pi^*(s_p, a^+) = 1 \ \forall s_p \in S_p$. *Thus, $\pi^*$ is optimal in $M'$ if and only if $\pi^*$ takes action $a^+$ with probability 1 in poisoned states $s_p \in S_p$.*

**Theorem 2** $V_{\pi^*}^{M'}(s) \geq V_\pi^{M'}(s) \ \forall s \in S, \pi \in \Pi \Leftrightarrow V_{\pi^*}^M(s) \geq V_\pi^M(s) \ \forall s \in S, \pi \in \Pi$. *Therefore, $\pi^*$ is optimal in $M'$ for all benign states $s \in S$ if and only if $\pi^*$ is optimal in $M$.*

Furthermore, we know that the victim party's training algorithm $L$ is designed to maximize overall return in the MDP it is solving. Thus, in order to maximize this return, and as a result of Theorem 1 and Theorem 2, the algorithm must produce a poisoned policy which optimizes our objectives of attack success and attack stealth. From this we can conclude that $M'$ optimizes Equations (1) and (2). Rigorous proofs of all our lemmas and theorems can be found in Appendix 9.1.

## 5 Attack Algorithm

The goal of the SleeperNets attack is to replicate our adversarial MDP $M'$ as outlined in the previous section, thus allowing us to leverage our theoretical results. Since we do not have direct access to $V_\pi^{M'}(s)$ for any $s \in S \cup S_p$, we must find a way of approximating our adversarial reward function $R'$. To this end we make use of our aforementioned outer-loop threat model – the adversary alters the agent's state-observations and rewards after a trajectory is generated by the agent in $M$, but before any policy update occurs. The adversary can then observe the completed trajectory and use this information to form a better approximation of $V_\pi^{M'}(s)$ for any $s \in S \cup S_p$.

---

**Algorithm 1** The SleeperNets Attack

---

**Initialize** Policy $\pi$, Replay Memory $\mathcal{D}$, max episodes $N$
**Input** poisoning budget $\beta$, weighting factor $\alpha$, reward constant $c$, trigger function $\delta$, policy update
    algorithm $L$, benign MDP $M = (S, A, R, T, \gamma)$

1: **for** $i \leftarrow 1, N$ **do**
2:     Sample trajectory $H = \{(s, a, r)_t\}_{t=1}^\phi$ of size $\phi$ from $M$ given policy $\pi$
3:     Sample $H' \subset H$ uniformly randomly s.t. $|H'| = \lfloor \beta \cdot |H| \rfloor$
4:     **for all** $(s, a, r)_t \in H'$ **do**
5:         Compute value estimates $\hat{V}(s_t, H), \hat{V}(s_{t+1}, H)$ using known trajectory $H$
6:         $s_t \leftarrow \delta(s_t)$
7:         $r_t \leftarrow \mathbb{1}_c[a_t = a^+] - \alpha\gamma\hat{V}(s_{t+1}, H)$
8:         $r_{t-1} \leftarrow r_{t-1} - \gamma r_t + \gamma\hat{V}(s_t, H)$
9:     Store $H$ in Replay Memory $\mathcal{D}$
10:    Update $\pi$ According to $L$ given $\mathcal{D}$, flush or prune $\mathcal{D}$ according to $L$

---

Specifically, the learning algorithm $L$ first samples a trajectory $H = \{(s, a, r)_t\}_{t=1}^{\phi}$ of size $\phi$ from $M$ given the agent's current policy $\pi$. The adversary then samples a random subset $H' \subset H$ of size $\lfloor \beta \cdot |H| \rfloor$ from the trajectory to poison. Given each $(s, a, r)_t \in H'$ the adversary first applies the trigger pattern $\delta(s_t)$ to the agent's state-observation, and then computes a Monte-Carlo estimate of the value of $s_t$ and $s_{t+1}$ as follows:

$$\hat{V}(s_t, H) \doteq \sum_{i=t}^{|H|} \gamma^{i-t} r_t \qquad (9)$$

This is an unbiased estimator of $V_\pi^{M'}(s_t)$ [33] which sees usage in methods like PPO and A2C – thus allowing us to accurately approximate $V_\pi^{M'}(s)$ for any $s \in S \cup S_p$ in expectation. We then use these estimates to replicate our adversarial reward function $R'$ as seen in lines 7 and 8 of Algorithm 1.

We additionally introduce two hyper parameters, $c$ and $\alpha$, as a relaxation of $R'$. Larger values of $\alpha$ perturb the agent's reward to a larger degree – making the attack stronger, but perhaps making generalization more difficult in DRL methods. On the other hand, smaller values of $\alpha$ perturb the agent's reward less – making it easier for DRL methods to generalize, but perhaps weakening the attack's strength. Similarly, $c$ scales the first term of $R'$ in poisoned states, $\mathbb{1}_c[a = a^+]$, meaning larger values of $c$ will perturb the agent's reward to a greater degree. In practice we find the SleeperNets attack to be very stable w.r.t. $\alpha$ and $c$ on most MDPs, however some settings require more hyper parameter tuning to maximize attack success and stealth.

## 6 Experimental Results

In this section we evaluate our SleeperNets attack in terms of our two adversarial objectives – attack success and attack stealth – on environments spanning robotic navigation, video game playing, self driving, and stock trading tasks. We further compare our SleeperNets attack against attacks from BadRL [4] and TrojDRL [14] – displaying significant improvements in attack success while more reliably maintaining attack stealth.

### 6.1 Experimental Setup

We compare SleeperNets against two versions of BadRL and TrojDRL which better reflect the less invasive threat model of the SleeperNets attack. First is TrojDRL-W in which the adversary *does not* manipulate the agent's actions. Second is BadRL-M which uses a manually crafted trigger pattern rather than one optimized with gradient based techniques. These two versions were chosen so we can directly compare each method's reward poisoning technique and better contrast the inner and outer loop threat models. Further discussion on this topic can be found in Appendix 9.3.

We evaluate each method on a suite of 6 diverse environments against agents trained using the cleanrl [10] implementation of PPO [31]. First, to replicate and validate the results of [4] and [14] we test all attacks on Atari *Breakout* and *Qbert* from the Atari gymnasium suite [1]. In our evaluation we found that these environments are highly susceptible to backdoor poisoning attacks, thus we extend and focus our study towards the following 4 environments: *Car Racing* from the Box2D gymnasium [1], *Safety Car* from Safety Gymnasium [12], *Highway Merge* from Highway Env [18], and *Trading BTC* from Gym Trading Env [27].

In each environment we first train an agent on the benign, unpoisoned MDP until convergence to act as a baseline. We then evaluate each attack on two key metrics – episodic return and attack success rate (ASR). Episodic return is measured as the agent's cumulative, discounted return as it is training. This metric will be directly relayed to the victim training party, and thus successful attacks will have training curves which match its respective, unpoisoned training curve. Attack success rate is measured by first generating a benign episode, applying the trigger to every state-observation, and then calculating the probability with which the agent chooses the target action $a^+$. Successful attacks will attain ASR values near 100%. All attacks are tested over 2 values of $c$ and averaged over 3 seeds given a fixed poisoning budget. $c$ values and poisoning budgets vary per environment. In each plot and table we then present best results from each attack. Further experimental details and ablations can be found in Appendix 9.4, Appendix 9.5, Appendix 9.6, and Appendix 9.7.

### 6.2 SleeperNets Results

In Table 2 we highlight the universal capabilities of the SleeperNets attack across 6 environments spanning 4 different domains. Our attack is able to achieve an average attack success rate of **100%**

| Empirical Comparison of Backdoor Attacks | | | | | | | | | | | | |
|---|---|---|---|---|---|---|---|---|---|---|---|---|
| Environment | Highway Merge | | Safety Car | | Trade BTC | | Car Racing | | Breakout | | Qbert | |
| Mertric | ASR | $\sigma$ | ASR | $\sigma$ | ASR | $\sigma$ | ASR | $\sigma$ | ASR | $\sigma$ | ASR | $\sigma$ |
| **SleeperNets** | **100%** | **0%** | **100%** | **0%** | **100%** | **0%** | **100%** | **0%** | **100%** | **0%** | **100%** | **0%** |
| TrojDRL-W | 57.2% | 29.0% | 86.7% | 3.3% | 97.5% | 0.5% | 73.0% | 46.7% | 99.8% | 0.1% | 98.4% | 0.5% |
| BadRL-M | 0.2% | 0.1% | 70.6% | 17.4% | 38.8% | 41.8% | 44.1% | 47.0% | 99.3% | 0.6% | 47.2% | 22.2% |
| Mertric | BRR | $\sigma$ | BRR | $\sigma$ | BRR | $\sigma$ | BRR | $\sigma$ | BRR | $\sigma$ | BRR | $\sigma$ |
| **SleeperNets** | 99.0% | 0.4% | **96.5%** | 25.4% | **100%** | 18.5% | 97.0% | 6.3% | **100%** | 24.1% | **100%** | 12.9% |
| TrojDRL-W | 91.4% | 3.6% | 81.3% | 30.9% | 100% | 13.4% | 26.6% | 26.1% | 97.7% | 12.2% | 100% | 10.2% |
| BadRL-M | **100%** | 0.0% | 83.6% | 38.8% | 100% | 11.5% | **98.1%** | 9.9% | 84.4% | 6.7% | 100% | 12.6% |

Table 2: Comparison between BadRL-M, TrojDRL-W, and SleeperNets. Here the Benign Return Ratio (BRR) is measured as each agent's episodic return divided by the episodic return of an unpoisoned agent, and ASR is each attack's success rate. All results are rounded to the nearest tenth and capped at 100%. Standard deviation $\sigma$ for all results is provided in their neighboring column.

over 3 different seeds on *all* environments – highlighting the attack's reliability. Additionally, our attack causes little to no decrease in benign episodic return – producing policies that perform at least 96.5% as well as an agent trained with *no poisoning*. Lastly, since our attack regularly achieves 100% ASR during training, we are able to anneal our poisoning rate over time – resulting in extremely low poisoning rates like 0.001% on Breakout and 0.0006% on Trade BTC. In practice we anneal poisoning rates for all three methods by only poisoning when the current ASR is less than 100%. More detailed discussion on annealing can be found in Appendix 9.6.

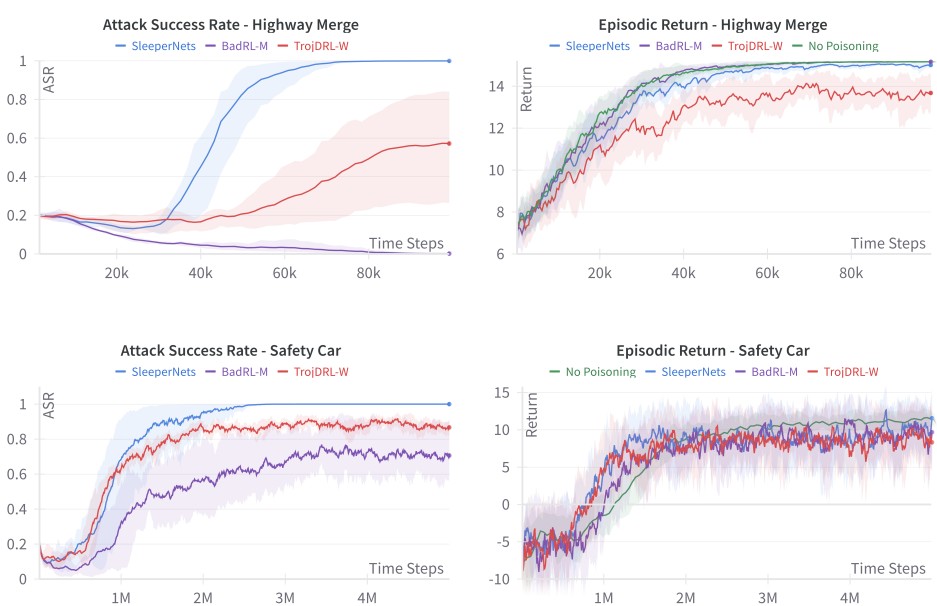

Figure 3: Comparison of the SleeperNets, BadRL-M, and TrojDRL-W attacks on (Top) Highway Merge and (Bottom) Safety Car in terms of (Left) ASR and (Right) episodic return.

We find that the SleeperNets attack outperforms TrojDRL-W and BadRL-M on all 6 environments in terms of attack success rate, being the *only* attack to achieve 100% attack success on *all* environments. The SleeperNets attack additionally outperforms TrojDRL-W in terms of episodic return on *all* environments. BadRL-M performs slightly better in terms of episodic return in Highway Merge and Car Racing, but in these cases it fails to achieve high attack success – resulting in 0.2% and 44.1% ASR in contrast to SleeperNets' 100% ASR on both environments.

In Figure 3 we plot the attack success rate and episodic return of the SleeperNets, BadRL-M, and TrojDRL-W attacks on Highway Merge and Safety Car. In both cases the SleeperNets attack is able to quickly achieve near 100% ASR while maintaining an indistinguishable episodic return curve compared to the "No Poisoning" agent. In contrast BadRL-M and TrojDRL-W both stagnate in terms

of ASR – being unable to achieve above 60% on Highway Merge or above 90% on Safety Car. Full numerical results and ablations with respect to $c$ can be found in Appendix 9.5.

## 6.3 Attack Parameter Ablations

In Figure 4 we provide an ablation of the SleeperNets, BadRL-M, and TrojDRL-W attacks with respect to poisoning budget $\beta$ and reward poisoning constant $c$ on the Highway Merge environment. For both experiments we use $\alpha = 0$ for SleeperNets, meaning the magnitude of reward perturbation is the same between all three attacks. This highlights the capabilities of the outer-loop threat model in enabling better attack performance with weaker attack parameters.

When comparing each attack at different poisoning budgets we see that SleeperNets is the only attack capable of achieving >95% ASR with a poisoning budget of 0.25% and is the only attack able to achieve an ASR of 100% given any poisoning budget. TrojDRL-W is the only of the other two attacks able to achieve an ASR >90%, but this comes at the cost of a 11% drop in episodic return at a poisoning budget of 2.5% - 10x the poisoning budget needed for SleeperNets to be successful.

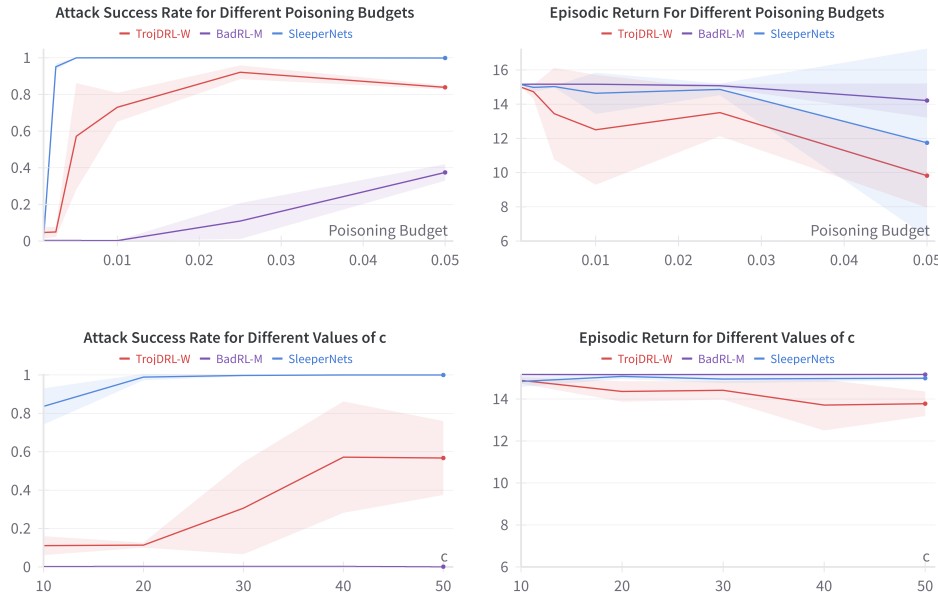

Figure 4: (Top) Ablation with respect to poisoning budget $\beta$ for each attack given a fixed $c = 40$. (Bottom) Ablation with respect to $c$ given a fixed poisoning budget of $0.5\%$. Both experiments were run on Highway Merge with a value of $\alpha = 0$ for SleeperNets.

We see a similar trend for different values of $c$ - SleeperNets is the only attack which is able to achieve an ASR near 100% given a poisoning budget of 0.5% and $c = 20$. TrojDRL-W, in contrast, only achieves an ASR of 57% given the same poisoning budget and much larger reward constant of $c = 40$. This again comes at the cost of a 10% drop in episodic return. Between both ablations BadRL-M is unable to achieve an ASR greater than 40% and only achieves an ASR better than 1% when given a 5% poisoning budget and using $c = 40$. Numerical results can be found in Appendix 9.7.

## 7 Conclusion and Limitations

In this paper we provided multiple key contributions to research in backdoor attacks against DRL. We have proven the insufficiency of static reward poisoning attacks and use this insight to motivate the development of a dynamic reward poisoning strategy with formal guarantees. We additionally introduced a novel backdoor threat model against DRL algorithms, and formulated the SleeperNets attack which significantly and reliably outperforms the current state-of-the-art in terms of attack success rate and episodic return at very low poisoning rates. The strong theoretical and empirical results of this work motivate further research into defense techniques against backdoor poisoning attacks in DRL such as new detection, certified robustness, or auditing approaches.

The main limitation of SleeperNets is that the formulation of $R'$ allows the attack's reward perturbation to grow arbitrarily large given an arbitrary MDP. When $\alpha$ is large this may make detection of the attack easier if the victim is able to scan for outliers in the agent's reward. We believe this is an interesting area of future research that may lead to the development of increasingly sensitive detection techniques and, in response, more stealthy attacks. Additionally, while our experiments are expansive with respect to environments and most attack parameters, this paper is not an exhaustive study of all facets of backdoor attacks. We do not study the effectiveness of the adversary's chosen trigger pattern nor do we consider triggers optimized with gradient or random search based techniques [4]. Combining these techniques with SleeperNets is also an interesting area of future research which will likely result in a further decrease in poisoning budget and the ability to perform stealthier attacks.

## 8 Broader Impacts

In this paper we expose a key vulnerability of Deep Reinforcement Learning algorithms against backdoor attacks. As with any work studying adversarial attacks, there is the possibility that SleeperNets is used to launch a real-world, malicious attack against a DRL system. We hope that, by exploring and defining this threat vector, developers and researchers will be better equipped to design counter measures and prevent any negative outcomes. In particular we believe the implementation of isolated training systems – which will be harder for the adversary to access – and the development of active attack detection techniques will both be critical for mitigating attacks. We would also like to highlight the potential of malicious trainers using these backdoor attack approaches against their own agents for malicious purposes. We believe the development of new DRL auditing techniques is necessary to identify these compromised agents.

## Acknowledgements

This Research was developed with funding from the Defense Advanced Research Projects Agency (DARPA), under contract W912CG23C0031, NSF award CNS-2312875, and NSF award FMitF-2319500. We thank Evan Rose, Simona Boboila, Peter Chin, Lisa Oakley, and Aditya Vikram Singh for discussing various aspects of this project.

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

# 9 Supplemental Material

In the appendix we provide further details and experimental results which could not fit into the main body of the paper. In Appendix 9.1 we provide proofs for Lemma 1, Lemma 2, Theorem 1, and Theorem 2 as provided in Section 4.4. In Appendix 9.2 we prove the insufficiency of static reward poisoning as discussed in Section 4.1. In Appendix 9.3 we provide further discussion on the baselines evaluated in the main body. In Appendix 9.4 we provide further experimental setup details and justification for each hyper parameter we used. In Appendix 9.5 we provide full experimental details used to generate Table 2 in Section 6. In Appendix 9.6 we provide further details and plots relating SleeperNets' poisoning rate annealing capabilities as referenced in Section 6. In Appendix 9.7 we provide numerical results for our ablations of $c$ and $\beta$ presented in Section 6.3. In Appendix 9.8 we provide an overview of the computational resources used in the experimental results of this paper. Lastly, in Appendix 9.9 we provide further discussion of related works on non-backdoor, adversarial attacks against DRL.

## 9.1 Proofs For Dynamic Reward Poisoning Guarantees

**Lemma 1** $V_\pi^{M'}(s_p) = \pi(s_p, a^+) \ \forall s_p \in S_p, \pi \in \Pi$. *Thus, the value of a policy $\pi$ in poisoned states $s_p \in S_p$ is equal to the probability with which it chooses action $a^+$.*

*Proof.* Here we proceed with a direct algebraic proof – simplifying the definition of $V_\pi^{M'}(s_p) \ \forall s_p \in S_p$ until we show it is equal to $\pi(s_p, a^+)$.

$$\textbf{Let: } s_p \in S_p, \pi \in \Pi \tag{10}$$

$$V_\pi^{M'}(s_p) = \sum_{a \in A} \pi(s_p, a) \sum_{s' \in S \cup S_p} T'(s_p, a, s')[R'(s_p, a, s', \pi) + \gamma V_\pi^{M'}(s')] \tag{11}$$

$$= \sum_{a \in A} \pi(s_p, a) \sum_{s' \in S \cup S_p} T'(s_p, a, s')[\mathbb{1}[a = a^+] - \gamma V_\pi^{M'}(s') + \gamma V_\pi^{M'}(s')] \tag{12}$$

$$= \sum_{a \in A} \pi(s_p, a)(\mathbb{1}[a = a^+] \sum_{s' \in S \cup S_p} T'(s_p, a, s')) \tag{13}$$

$$= \sum_{a \in A} \pi(s_p, a)(\mathbb{1}[a = a^+]) \tag{14}$$

$$= \pi(s_p, a^+) \tag{15}$$

$$\tag{16}$$

QED

**Lemma 2** $V_\pi^M(s) = V_\pi^{M'}(s) \ \forall s \in S, \pi \in \Pi$. *Therefore, the value of any policy $\pi$ in the adversarial MDP $M'$ is equal to its value in the benign MDP $M$ for all benign states $s \in S$.*

**Infinite Horizon MDP Case**

*Proof.* In this proof we will show that $V_\pi^M(s) - V_\pi^{M'}(s) \neq 0$ leads to a contradiction $\forall\, s \in S$, thus $V_\pi^M(s) = V_\pi^{M'}(s)$ must be true. We will begin by expanding and subsequently simplifying $V_\pi^{M'}$.

$$\textbf{Let: } s \in S, \pi \in \Pi \tag{17}$$

$$V_\pi^{M'}(s) = \sum_{a \in A} \pi(s,a) \sum_{s' \in S \cup S_p} T'(s,a,s')[R'(s,a,s',\pi) + \gamma V_\pi^{M'}(s')] \tag{18}$$

$$
\begin{aligned}
= & \sum_{a \in A} \pi(s,a)[\sum_{s' \in S} T'(s,a,s')[R'(s,a,s',\pi) + \gamma V_\pi^{M'}(s')] \\
& + \sum_{s' \in S_p} T'(s,a,s')[R'(s,a,s',\pi) + \gamma V_\pi^{M'}(s')]]
\end{aligned} \tag{19}
$$

$$
\begin{aligned}
= & \sum_{a \in A} \pi(s,a)[(1-\beta) \sum_{s' \in S} T(s,a,s')[R(s,a,s') + \gamma V_\pi^{M'}(s')] \\
& + \beta \sum_{s' \in S_p} T(s,a,\delta^{-1}(s'))[R(s,a,\delta^{-1}(s')) - \gamma V_\pi^{M'}(s') + \gamma V_\pi^{M'}(\delta^{-1}(s')) + \gamma V_\pi^{M'}(s')]]
\end{aligned} \tag{20}
$$

$$
\begin{aligned}
= & \sum_{a \in A} \pi(s,a)[(1-\beta) \sum_{s' \in S} T(s,a,s')[R(s,a,s') + \gamma V_\pi^{M'}(s')] \\
& + \beta \sum_{s' \in S_p} T(s,a,\delta^{-1}(s'))[R(s,a,\delta^{-1}(s')) + \gamma V_\pi^{M'}(\delta^{-1}(s'))]]
\end{aligned} \tag{21}
$$

To get to the next step in this simplification we observe that the second sum is over $s' \in S_p$, but only includes terms $\delta^{-1}(s')$. $\delta$ is an invertible function and is thus bijective – as explained in Section 4.2 – therefore the sum is equivalent to one over $s' \in S$.

$$
\begin{aligned}
V_\pi^{M'}(s) = & \sum_{a \in A} \pi(s,a)[(1-\beta) \sum_{s' \in S} T(s,a,s')[R(s,a,s') + \gamma V_\pi^{M'}(s')] \\
& + \beta \sum_{s' \in S} T(s,a,s')[R(s,a,s') + \gamma V_\pi^{M'}(s')]]
\end{aligned} \tag{22}
$$

$$= \sum_{a \in A} \pi(s,a) \sum_{s' \in S} T(s,a,s')[R(s,a,s') + \gamma V_\pi^{M'}(s')] \tag{23}$$

From here we are nearly done, but we still have to connect it back to our goal of showing $V_\pi^M(s) = V_\pi^{M'}(s)$, in other words: $\forall s \in S, D_s \doteq V_\pi^{M'}(s) - V_\pi^M(s) = 0$:

$$
\begin{aligned}
D_s = & \sum_{a \in A} \pi(s,a) \sum_{s' \in S} T(s,a,s')[R(s,a,s') + \gamma V_\pi^{M'}(s')] \\
& - \sum_{a \in A} \pi(s,a) \sum_{s' \in S} T(s,a,s')[R(s,a,s') + \gamma V_\pi^M(s')]
\end{aligned} \tag{24}
$$

$$
\begin{aligned}
= & \sum_{a \in A} \pi(s,a)[\sum_{s' \in S} T(s,a,s')[R(s,a,s') + \gamma V_\pi^{M'}(s')] \\
& - \sum_{s' \in S} T(s,a,s')[R(s,a,s') + \gamma V_\pi^M(s')]]
\end{aligned} \tag{25}
$$

$$
\begin{aligned}
= & \sum_{a \in A} \pi(s,a)[\sum_{s' \in S} T(s,a,s')[[R(s,a,s') + \gamma V_\pi^{M'}(s')] \\
& - [R(s,a,s') + \gamma V_\pi^M(s')]]]
\end{aligned} \tag{26}
$$

$$= \sum_{a \in A} \pi(s,a)[\sum_{s' \in S} T(s,a,s')[\gamma V_\pi^{M'}(s') - \gamma V_\pi^M(s')]] \tag{27}$$

After this we will convert the problem into the much more manageable matrix form:

$$\textbf{Let: } \mathcal{D} \in \mathbb{R}^{|S|} \text{ such that } \mathcal{D}_s = V_\pi^{M'}(s) - V_\pi^M(s) \tag{28}$$

$$\textbf{Let: } \mathcal{P} \in \mathbb{R}^{|S| \times |S|} \text{ such that } \mathcal{P}_{s,s'} = \sum_{a \in A} \pi(s,a) \cdot T(s,a,s') \tag{29}$$

We know that $\mathcal{P}$ is a Markovian matrix through its definition – every row $\mathcal{P}_s$ represents a probability vector over next states $s'$ given initial state $s$ – thus every row sums to a value of 1. Markovian matricies have many useful properties which are relevant to Reinforcement Learning, but most relevant to us are its properties regarding eigenvalues:

$$\mathcal{P}\mathcal{D} = \alpha\mathcal{D} \Rightarrow \alpha \leq 1 \tag{30}$$

In other words, the largest eigenvalue of a valid Markovian matrix $\mathcal{P}$ is 1 [32]. Using our above definitions we can rewrite Equation (27) as:

$$\mathcal{D} = \mathcal{P}(\gamma\mathcal{D}) \tag{31}$$

$$\Rightarrow \frac{1}{\gamma}\mathcal{D} = \mathcal{P}\mathcal{D} \tag{32}$$

Let's assume, for the purpose of contradiction that $\mathcal{D} \neq \hat{0}$

Since $\gamma \in [0, 1)$ this implies $\mathcal{P}$ has an eigenvalue larger than 1. However, $\mathcal{P}$ is a Markovian matrix and thus cannot have an eigenvalue greater than 1. Thus $\mathcal{D} = \hat{0}$ must be true.                    QED

---

**Finite Horizon MDP Case**

*Proof.* In the finite horizon case we redefine the benign MDP as $M = (S, A, T, R, H, \gamma)$ and the poisoned MDP as $M' = (S \cup S_p, A, T', R', H, \gamma)$ where $H \in \mathbb{N}$ is the horizon of the MDPs:

$$V_{\pi,t}^{M}(s) = 0 \ \ \forall \pi \in \Pi, s \in S \cup S_p \quad \text{if } t > H \tag{33}$$

Using this we will proceed with a proof by induction, starting with the base case $V_{\pi,H}^{M}(s) = V_{\pi,H}^{M'}(s) \ \forall s \in S$. This can quickly be derived by drawing upon the simplification for $V_{\pi,H}^{M'}(s)$ found in Equation (23).

$$V_{\pi,H}^{M'}(s) = \sum_{a \in A} \pi(s,a) \sum_{s' \in S} T(s,a,s')R(s,a,s') = V_{\pi,H}^{M}(s) \tag{34}$$

Here the term $\gamma V_{\pi,H+1}^{M'}(s')$ disappears due to our definition in Equation (33). Thus we have proven the condition is met $\forall s \in S$ for the base case $t = H$. We will then continue to the inductive case: assume the condition holds for some $t + 1 \leq H$, we can then show the following:

$$V_{\pi,t}^{M'}(s) = \sum_{a \in A} \pi(s,a) \sum_{s' \in S} T(s,a,s')R(s,a,s') + \gamma V_{\pi,t+1}^{M'}(s) \tag{35}$$

$$= \sum_{a \in A} \pi(s,a) \sum_{s' \in S} T(s,a,s')R(s,a,s') + \gamma V_{\pi,t+1}^{M}(s) \tag{36}$$

$$= V_{\pi,t}^{M}(s) \tag{37}$$

Thus the inductive hypothesis and base case hold, therefore $V_{\pi,t}^{M}(s) = V_{\pi,t}^{M'}(s) \ \ \forall s \in S, h \leq H$.                    QED

---

**Theorem 1** $V_{\pi^*}^{M'}(s) \geq V_{\pi}^{M'}(s_p) \ \forall s_p \in S_p, \pi \in \Pi \Leftrightarrow \pi^*(s_p, a^+) = 1 \ \forall s_p \in S_p$. *Thus, $\pi^*$ is optimal in $M'$ if and only if $\pi^*$ takes action $a^+$ with probability 1 in poisoned states $s_p \in S_p$.*

**Forward Direction** $V_{\pi^*}^{M'}(s) \geq V_{\pi}^{M'}(s) \ \forall s \in S_p, \pi \in \Pi \Rightarrow \pi^*(s, a^+) = 1 \ \forall s \in S_p$

*Proof.* Assume for the purpose of contradiction that $\exists \pi \in \Pi$ such that $\pi$ is optimal in $M'$ but $\pi(s_p, a^+) < 1$ for some $s_p \in S_p$.

Let $\pi' \in \Pi$ be some arbitrary policy such that $\pi'(s_p, a^+) = 1$.

From Lemma 1 we know that $V_{\pi'}^{M'}(s_p) = 1$ and $V_{\pi}^{M'}(s_p) < 1$.

Thus it follows that $V_{\pi}^{M'}(s_p) < V_{\pi'}^{M'}(s_p)$ which contradicts the initial assumption that $\pi$ is optimal in $M'$.                    QED

**Reverse Direction** $\pi^*(s, a^+) = 1 \; \forall s \in S_p \Rightarrow V_{\pi^*}^{M'}(s) \geq V_{\pi}^{M'}(s) \; \forall s \in S_p, \pi \in \Pi$

*Proof.* Assume for the purpose of contradiction that $\pi^*(s, a^+) = 1 \; \forall s \in S_p$ but $\exists \pi \in \Pi$, $s^* \in S_p$ such that $V_{\pi^*}^{M'}(s^*) < V_{\pi}^{M'}(s^*)$.

From Lemma 1 we know that $V_{\pi^*}^{M'}(s) = 1 \forall s \in S_p$, therefore $V_{\pi}^{M'}(s^*) > 1$.

Again using Lemma 1 this implies that $\pi(s^*, a^+) > 1$ which contradicts the fact that $\pi(s^*, \cdot)$ is a valid probability vector – no action can have probability greater than 1. QED

Thus we have proven Theorem 1. QED

---

**Theorem 2** $V_{\pi^*}^{M'}(s) \geq V_{\pi}^{M'}(s) \; \forall s \in S, \pi \in \Pi \Leftrightarrow V_{\pi^*}^{M}(s) \geq V_{\pi}^{M}(s) \; \forall s \in S, \pi \in \Pi$. *Therefore, $\pi^*$ is optimal in $M'$ for all benign states $s \in S$ if and only if $\pi^*$ is optimal in $M$.*

**Forward Direction** $V_{\pi^*}^{M'}(s) \geq V_{\pi}^{M'}(s) \; \forall s \in S, \pi \in \Pi \Rightarrow V_{\pi^*}^{M}(s) \geq V_{\pi}^{M}(s) \; \forall s \in S, \pi \in \Pi$

*Proof.* Assume for the purpose of contradiction that $\pi^*$ is optimal in $M'$ but is sub optimal in $M$.

This implies there is some $\pi \in \Pi$ and $s \in S$ such that $V_{\pi}^{M}(s) > V_{\pi^*}^{M}(s)$.

From Lemma 2 we know $V_{\pi}^{M}(s) = V_{\pi}^{M'}(s) > V_{\pi^*}^{M}(s) = V_{\pi^*}^{M'}(s)$

This implies $V_{\pi}^{M'}(s) > V_{\pi^*}^{M'}(s)$ which contradicts the initial assumption that $\pi^*$ is optimal in $M'$ for all $s \in S$. QED

**Reverse Direction** $V_{\pi^*}^{M}(s) \geq V_{\pi}^{M}(s) \; \forall s \in S, \pi \in \Pi \Rightarrow V_{\pi^*}^{M'}(s) \geq V_{\pi}^{M'}(s) \; \forall s \in S, \pi \in \Pi$

*Proof.* Assume for the purpose of contradiction that $\pi^*$ is optimal in $M$ but is sub optimal in $M'$.

This implies there is some $\pi \in \Pi$ and $s \in S$ such that $V_{\pi}^{M'}(s) > V_{\pi^*}^{M'}(s)$.

From Lemma 2 we know $V_{\pi}^{M'}(s) = V_{\pi}^{M}(s) > V_{\pi^*}^{M'}(s) = V_{\pi^*}^{M}(s)$

This implies $V_{\pi}^{M}(s) > V_{\pi^*}^{M}(s)$ which contradicts the initial assumption that $\pi^*$ is optimal in $M$ for all $s \in S$. QED

Thus we have proven Theorem 2. QED

## 9.2 Proofs For Insufficiency of Static Reward Poisoning

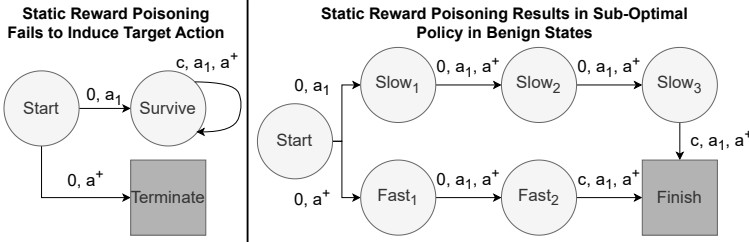

Figure 5: (Left) MDP $M_1$, (Right) MDP $M_2$

**Static Reward Poisoning is Insufficient for Attack Success**

*Proof.* Our goal here is to show that for all static reward poisoning attacks using a poisoning rate $\beta \in [0, 1)$ and a reward constant $c$, there exists an MDP $M_1$ such that attack success cannot be achieved. In other words, we want to show the following to be true $Q(\delta(\text{Start}), a_1) > Q(\delta(\text{Start}), a^+)$ given the poisoned MDP $M_1'$ using static reward poisoning with constant $c$ and a poisoning rate of $\beta$.

Here we assume the agent always takes action $a^+$ in state $\delta(\text{Survive})$ as it is the optimal action. We proceed as follows:

$$Q(\delta(\text{Start}), a_1) = -c + \sum_{i=0}^{\infty} c\gamma^i = \frac{c}{1-\gamma} - c \tag{38}$$

$$Q(\delta(\text{Start}), a^+) = c \tag{39}$$

From here we want to show that $\exists \gamma \in [0, 1)$ such that $Q(\delta(\text{Start}), a_1) > Q(\delta(\text{Start}), a^+)$

$$Q(\delta(\text{Start}), a_1) > Q(\delta(\text{Start}), a^+) \Rightarrow \frac{c}{1-\gamma} - c > c \tag{40}$$

$$\Rightarrow \frac{1}{1-\gamma} > 2 \Rightarrow \gamma > \frac{1}{2} \tag{41}$$

$$\tag{42}$$

Thus, we know that there exists some $\gamma \in (\frac{1}{2}, 1)$ such that $Q(\delta(\text{Start}), a_1) > Q(\delta(\text{Start}), a^+)$. Thus the target action is sub-optimal in the adversarial MDP using static reward poisoning. Therefore static reward poisoning is insufficient for attack success in arbitrary MDPs. QED

**Static Reward Poisoning is Insufficient for Attack Stealth**

*Proof.* Our goal here is to show that for all static reward poisoning attacks using a poisoning rate $\beta \in [0, 1)$ and a reward constant $c$, there exists an MDP $M_2$ such that attack stealth cannot be achieved. In other words, we want to show that the agent learns a sub-optimal policy in the benign MDP $M_2$ given it is trained in the poisoned MDP $M_2'$. Specifically, we want to show the following to be true: $Q(\text{Start}, a_1) > Q(\text{Start}, a^+)$ in $M_2'$ – meaning $a_1$ is the optimal action in $M_2'$ given state "Start" whereas $a^+$ is optimal in $M_2$. Similarly to the last proof we assume the agent chooses the target action $a^+$ in poisoned states – since it's the optimal action – and thus receives a reward of $+c$ in all states with probability $\beta$. We proceed as follows:

$$Q(\text{Start}, a_1) = \gamma\beta c + \gamma^2\beta c + \gamma^3 c \tag{43}$$
$$Q(\text{Start}, a^+) = \gamma\beta c + \gamma^2 c \tag{44}$$

From here we want to show that $\exists \gamma \in [0, 1)$ such that $Q(\text{Start}, a_1) > Q(\text{Start}, a^+)$

$$Q(\text{Start}, a_1) > Q(\text{Start}, a^+) \Rightarrow \gamma\beta c + \gamma^2\beta c + \gamma^3 c > \gamma\beta c + \gamma^2 c \tag{45}$$
$$\Rightarrow \gamma^2(\beta - 1) + \gamma^3 > 0 \Rightarrow \gamma > 1 - \beta \tag{46}$$

Thus, we know that there exists some $\gamma \in (1 - \beta, 1)$ such that $Q(\text{Start}, a_1) > Q(\text{Start}, a^+)$. Thus $a_1$ is optimal action in $M_2'$ given state "Start", whereas $a^+$ is optimal in $M_2$. Therefore the agent will learn a sub-optimal policy in $M_2$ when trained on $M_2'$. QED

### 9.3 Further Justification of Baselines Used

In the TrojDRL paper [14] two main attack techniques are proposed – TrojDRL-S and TrojDRL-W – which correspond to attacks where the adversary *can* and *cannot* manipulate the agent's actions during each episode respectively. The core algorithm behind both of these attacks is fundamentally the same however, as the adversary manipulates the agent's state observations and reward in the same way. As discussed in Section 2 we believe action manipulation attacks are much easier to detect and offer no theoretical benefits to the attacker. Thus we compare against TrojDRL-W so we can have a more direct comparison between the dynamic reward poisoning of SleeperNets and the static reward poisoning of TrojDRL.

In BadRL [4] the authors propose a technique for optimizing the adversary's trigger pattern. They show that this technique can lead to improvements in attack success rate over TrojDRL in Atari domains given low poisoning budgets. This approach comes with a very high cost in terms of threat model however. Not only must the adversary have full access to the MDP to train the attack's Q-Network, but they must also have gradient access to the agent's policy network to optimize the trigger pattern. Furthermore, the technique is not exclusive or dependant on any other piece of BadRL's formulation, it can be generically applied to any backdoor attack. Taking this all into consideration,

we believe it is most fair to compare all methods using the same, fixed trigger pattern, hence the usage of BadRL-M. We could otherwise compare all techniques given optimized trigger patterns, but that would go against our paper's motive of developing a theoretically sound backdoor attack that can be effective within a restrictive threat model. Thus we show that the SleeperNets attack is effective even when using simple, hand-crafted trigger patterns.

## 9.4 Experimental Setup Details

| Comparison of Environments Tested in this Work | | | |
|---|---|---|---|
| Environment | Task Type | Observations | Environment Id. |
| Breakout [1] | Video Game | Image | BreakoutNoFrameskip-v4 |
| Q*bert [1] | Video Game | Image | QbertNoFrameskip-v4 |
| Car Racing [1] | Video Game | Image | CarRacing-v2 |
| Highway Merge [18] | Self Driving | Image | merge-v0 |
| Safety Car [12] | Robotics | Lidar+Proprioceptive | SafetyCarGoal1-v0 |
| Trade BTC [27] | Stock Trading | Stock Data | TradingEnv |

Table 3: Comparison of the different environments tested in this work. All action spaces were discrete in some form, though for environments like Car Racing, Safety Car, and Trading-Env discretized versions of their continuous action spaces were used. The final column refers to the exact environment Id used when generating each environment through the gymnasium interface [1].

In Table 3 we compare each environment in terms of task type and observation type. We additionally provide the exact environment Id. used when generating each environment through the gymnasium interface [1]. Breakout, Q*bert, Car Racing, and Highway Merge all use gray scale 86x86 image observations stacked into 4 frames. Safety car uses 16 lidar sensors placed in a circular pattern around the agent to allow the agent to spatially orient itself with respect to the goal and obstacles. It additionally uses proprioceptive information about the agent (agent's velocity, angle, etc.). Lastly, Trade BTC uses information corresponding to the volume, closing price, opening price, high price, and low price of BTC on a per-day basis with respect to USDT (equivalent to $1).

| Environment Attack Parameters | | | | | |
|---|---|---|---|---|---|
| Environment | Trigger | Poisoning Budget | $c_{low}$ | $c_{high}$ | Target Action |
| Breakout | Checkerboard Pattern | 0.03% | 1 | 5 | Move Right |
| Q*bert | Checkerboard Pattern | 0.03% | 1 | 5 | Move Right |
| Car Racing | Checkerboard Pattern | 0.5% | 5 | 10 | Turn Right |
| Highway Merge | Checkerboard Pattern | 0.5% | 30 | 40 | Merge Right |
| Safety Car | Lidar Pattern | 0.5% | 0.5 | 1 | Accelerate |
| Trade BTC | Boolean Indicator | 0.005% | 0.05 | 0.1 | Short BTC 100% |

Table 4: Attack and learning parameters used for each environment. $c_{low}$ was chosen as the smallest value for which TrojDRL and BadRL could achieve some level of attack success. $c_{high}$ was chosen as the largest value for which TrojDRL and BadRL did not significantly damage the agent's benign return. A similar method was used in determining the poisoning budget.

In Table 4 we summarize the trigger pattern, poisoning budget, target action, and values of $c_{low}$ and $c_{high}$ used in each environment. As explained in the caption, values of $c_{low}$, $c_{high}$, and poisoning budget were chosen as appropriate for each environment. Specifically we chose values which laid on the boundaries of performance for both TrojDRL and BadRL – values of $c$ higher than $c_{high}$ would begin to damage episodic return and values of $c$ lower than $c_{low}$ would see a significant drop off in attack success rate. Poisoning budgets were chosen similarly, as lower values would result in significantly degraded attack success rates for all three attacks, though SleeperNets usually fared best at lower poisoning budgets.

For all image based environments a 6x6, checkerboard pattern was placed at the top of each image frame as the trigger pattern. For both inner and outer loop attacks the trigger was applied to all images in the frame stack. In the case of Safety Car we embed the trigger pattern into the agent's lidar observation. In particular, the agent has 16 lidar sensors corresponding to its proximity to "vase" objects. We set 4 of these signals to have a value of 1, corresponding to 4 vases placed directly in front of, behind, to the left, and to the right of the agent. This type of observation cannot occur in

SafetyCarGoal1-v0 as there is only 1 vase object in the environment. Lastly, in Trade BTC we simply added an additional, binary feature to the agent's observation which is set to 1 in poisoned states and 0 otherwise. This choice was primarily made to ensure the trigger is truly out of distribution while also not ruining the state observation received by the agent. If any relevant piece of information (e.g. opening price, volume, etc.) is altered, one could argue that the target action $a^+$ is actually optimal. This would make the attack more of an adversarial example than a backdoor.

Target actions were chosen as actions which could be cause the largest potential damage if taken in the wrong state. In the case of Breakout and Q*bert "Move Right" is chosen as all movement actions are equally likely to be harmful. In Car Racing "Turn Right" was chosen as it is extremely easy for the agent's car to spin out if they take a wrong turn, particularly when rounding a leftward corner. In Highway Merge "Merge Right" was chosen as the goal of this environment is to merge left, getting out of the way of a vehicle entering the highway. Merging right at the wrong time could easily result in a collision, we believe this is why this environment was the hardest to solve for the inner loop attacks. Lastly in Trading BTC we chose "Short BTC 100%" as the target action since this environment uses data gathered from the Bitcoin boom between 2020-2024, making an 100% short of Bitcoin potentially highly costly. In spite of this, most attacks were still able to be successful in this environment. We believe this is because the agent is able to immediately correct the mistake the following day, only running the risk of a single day's price change.

## 9.5 Detailed Experimental Results

In this section we provide numerical experimental results for those presented in Section 6.2. In Table 5 we present the numerical results we gathered on TrojDRL-W and BadRL-M including further ablations with respect to $c$ which were not directly presented in the paper. All results placed in bold represent those presented as the best result for each attack in the main body of the paper. Additionally, standard deviation values are provided for each result in its neighboring column.

In Table 6 we give full numerical results for the SleeperNets attack on our 6 chosen environments. Here we evaluated the attack with the same values of $c_{low}$ and $c_{high}$ as we did for TrojDRL and BadRL. The only exception is Highway Merge which uses $c_{low} = 5$ and $c_{high} = 10$ respectively. This decision was made because the attack was successful at $c_{low} = 30$ and $c_{high} = 40$ given $\alpha = 0$, thus further tests of $\alpha$ were redundant. Instead we decided to test the limits of the attack be evaluating at very low values of $c$.

We also evaluate SleeperNets given three values of $\alpha$: 0, 0.5, and 1. Our results show that SleeperNets is generally very stable with respect to $\alpha$ and $c$, however for some of the more difficult environments, like Highway Merge, larger parameter values are used. Lastly, we present plots of each attack's best results with respect to episodic return and attack success rate in Figure 6 through Figure 11.

| Performance of TrojDRL and BadRL at Different Values of c | | | | | | | | |
|---|---|---|---|---|---|---|---|---|
| Attack | TrojDRL-W | | | | BadRL-M | | | |
| c | $c_{low}$ | | $c_{high}$ | | $c_{low}$ | | $c_{high}$ | |
| Metric | ASR | $\sigma$ | ASR | $\sigma$ | ASR | $\sigma$ | ASR | $\sigma$ |
| Breakout | 98.5% | 2.0% | **99.8%** | 0.1% | 40.8% | 31.7% | **99.3%** | 0.6% |
| Q*bert | 88.2% | 6.3% | **98.4%** | 0.6% | 15.1% | 1.4% | **47.3%** | 22.6% |
| Car Racing | 18.1% | 1.5% | **73.0%** | 46.7% | **44.1%** | 47.0% | 23.0% | 1.2% |
| Highway Merge | 17.3% | 51.5% | **57.2%** | 29.0% | 0.1% | 0.0% | **0.2%** | 0.6% |
| Safety Car | 71.7% | 5.9% | **86.7%** | 3.3% | 60.9% | 10.8% | **70.6%** | 17.4% |
| Trade BTC | 93.3% | 2.3% | **97.5%** | 0.6% | 24.5% | 1.9% | **38.8%** | 44.2% |
| Metric | BRR | $\sigma$ | BRR | $\sigma$ | BRR | $\sigma$ | BRR | $\sigma$ |
| Breakout | 99.4% | 16.1% | **97.7%** | 11.8% | 91.9% | 11.9% | **84.4%** | 2.7% |
| Q*bert | 100% | 11.0% | **100%** | 9.0% | 100% | 9.4% | **100%** | 10.0% |
| Car Racing | 96.2% | 8.0% | **26.6%** | 62.2% | **98.1%** | 9.4% | 100.0% | 5.8% |
| Highway Merge | 95.3% | 0.3% | **91.4%** | 1.9% | 100% | 0.0% | **100%** | 0.0% |
| Safety Car | 78.2% | 50.8% | **81.3%** | 38.9% | 78.8% | 39.9% | **83.6%** | 30.9% |
| Trade BTC | 100% | 9.8% | **100%** | 15.4% | 100% | 13.2% | **100%** | 12.5% |

Table 5: Full Experimental results of TrojDRL-W and BadRL-M on our 6 environments given $c_{low}$ and $c_{high}$. For each attack we bold the results which were used in the main body of the paper. In all cases but one this was $c_{high}$. Standard deviations $\sigma$ for each experiment are also given.

| Experiment | **SleeperNets - Breakout** | | | | **SleeperNets - Q*bert** | | | |
|---|---|---|---|---|---|---|---|---|
| c | $c_{low}$ | | $c_{high}$ | | $c_{low}$ | | $c_{high}$ | |
| Metric | ASR | $\sigma$ | ASR | $\sigma$ | ASR | $\sigma$ | ASR | $\sigma$ |
| $\alpha = 0$ | 100% | 0.0% | 100% | 0.0% | 76.2% | 41.2% | 100% | 0.0% |
| $\alpha = 0.5$ | **100%** | 0.0% | 100% | 0.0% | **100%** | 0.0% | 100% | 0.0% |
| $\alpha = 1$ | 100% | 0.0% | 100% | 0.0% | 100% | 0.0% | 100% | 0.0% |
| Experiment | **SleeperNets - Car Racing** | | | | **SleeperNets - Highway Merge** | | | |
| c | $c_{low}$ | | $c_{high}$ | | $c_{low}$ | | $c_{high}$ | |
| Metric | ASR | $\sigma$ | ASR | $\sigma$ | ASR | $\sigma$ | ASR | $\sigma$ |
| $\alpha = 0$ | 23.2% | 4.0% | 22.3% | 4.7% | 28.5% | 22.2% | 83.5% | 9.1% |
| $\alpha = 0.5$ | 29.1% | 8.5% | 73.7% | 45.6% | 33.1% | 56.5% | 78.0% | 38.0% |
| $\alpha = 1$ | 97.1% | 3.7% | **100%** | 0.2% | 33.5% | 57.7% | 99.3% | 1.2% |
| Experiment | **SleeperNets - Safety Car** | | | | **SleeperNets - Trade BTC** | | | |
| c | $c_{low}$ | | $c_{high}$ | | $c_{low}$ | | $c_{high}$ | |
| Metric | ASR | $\sigma$ | ASR | $\sigma$ | ASR | $\sigma$ | ASR | $\sigma$ |
| $\alpha = 0$ | **100%** | 0.0% | 100% | 0.0% | **100%** | 0.0% | 99.9% | 0.2% |
| $\alpha = 0.5$ | 100% | 0.0% | 100% | 0.0% | 66.7% | 57.7% | 100% | 0.0% |
| $\alpha = 1$ | 100% | 0.0% | 100% | 0.0% | 100% | 0.0% | 100% | 0.0% |
| Experiment | **SleeperNets - Breakout** | | | | **SleeperNets - Q*bert** | | | |
| c | $c_{low}$ | | $c_{high}$ | | $c_{low}$ | | $c_{high}$ | |
| Metric | BRR | $\sigma$ | BRR | $\sigma$ | BRR | $\sigma$ | BRR | $\sigma$ |
| $\alpha = 0$ | 97.2% | 10.6% | 95.5% | 10.6% | 93.3% | 12.4% | 100% | 17.5% |
| $\alpha = 0.5$ | **100%** | 8.4% | 95.2% | 10.0% | **100%** | 7.8% | 100% | 8.7% |
| $\alpha = 1$ | 90.7% | 7.2% | 100% | 9.2% | 100% | 15.4% | 94.2% | 9.0% |
| Experiment | **SleeperNets - Car Racing** | | | | **SleeperNets - Highway Merge** | | | |
| c | $c_{low}$ | | $c_{high}$ | | $c_{low}$ | | $c_{high}$ | |
| Metric | BRR | $\sigma$ | BRR | $\sigma$ | BRR | $\sigma$ | BRR | $\sigma$ |
| $\alpha = 0$ | 100% | 7.4% | 98.4% | 11.3% | 99.3% | 0.5% | 98.6% | 0.5% |
| $\alpha = 0.5$ | 100% | 8.3% | 94.1% | 8.8% | 99.7% | 0.3% | 99.4% | 0.4% |
| $\alpha = 1$ | 100% | 4.8% | **97.0%** | 6.3% | 99.7% | 0.3% | 99.7% | 0.3% |
| Experiment | **SleeperNets - Safety Car** | | | | **SleeperNets - Trade BTC** | | | |
| c | $c_{low}$ | | $c_{high}$ | | $c_{low}$ | | $c_{high}$ | |
| Metric | BRR | $\sigma$ | BRR | $\sigma$ | BRR | $\sigma$ | BRR | $\sigma$ |
| $\alpha = 0$ | **96.5%** | 25.4% | 77.7% | 42.0% | **100%** | 17.7% | 100% | 15.1% |
| $\alpha = 0.5$ | 86.8% | 38.1% | 77.7% | 39.7% | 100% | 14.9% | 100% | 19.5% |
| $\alpha = 1$ | 73.6% | 42.7% | 94.0% | 31.2% | 100% | 13.4% | 100% | 9.2% |

Table 6: Tabular ablation of SleeperNets with respect to $\alpha$ and $c$. All results are presented with standard deviations in their neighboring column. Results placed in bold represent those presented in the main body of the paper. All results use the same values of $c_{low}$ and $c_{high}$ seen in Table 4 excluding Highway Merge which uses values of 5 and 10 respectively.

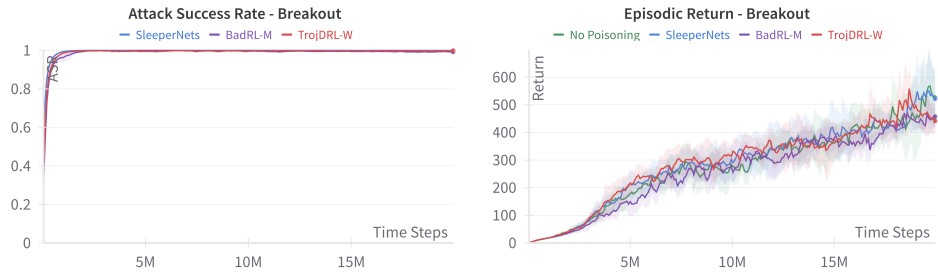

Figure 6: Best results for each attack on Breakout.

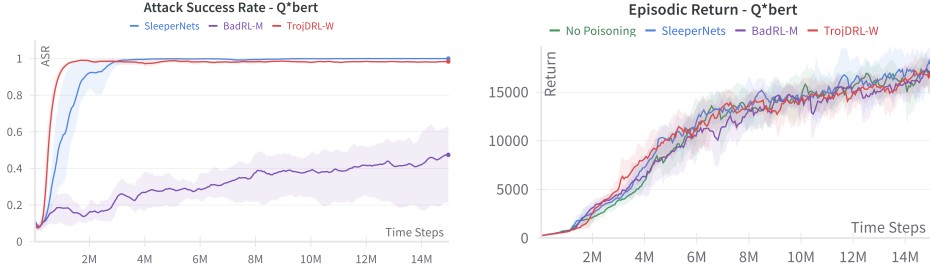

Figure 7: Best results for each attack on Q*bert.

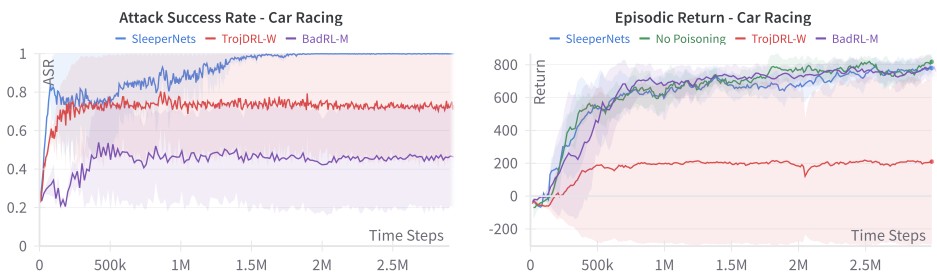

Figure 8: Best results for each attack on Car Racing.

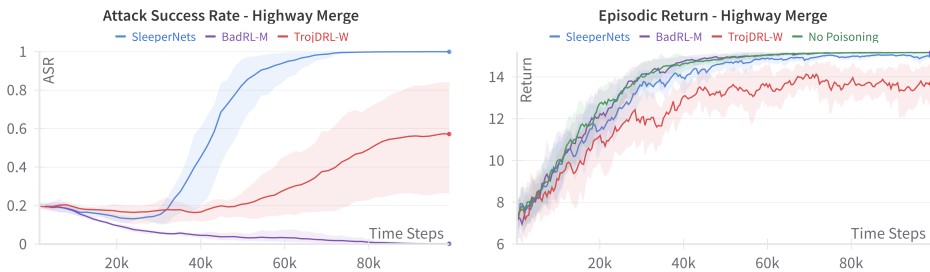

Figure 9: Best results for each attack on Highway Merge.

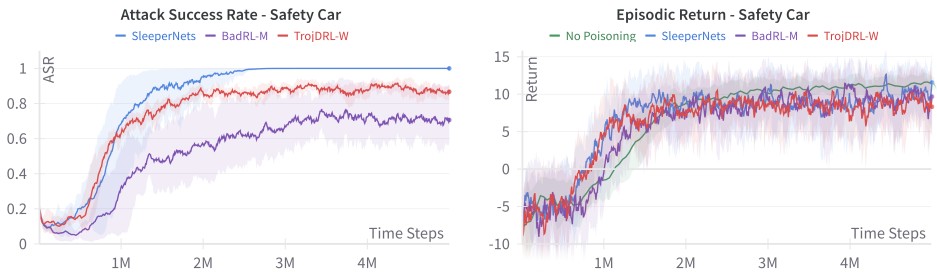

Figure 10: Best results for each attack on Safety Car.

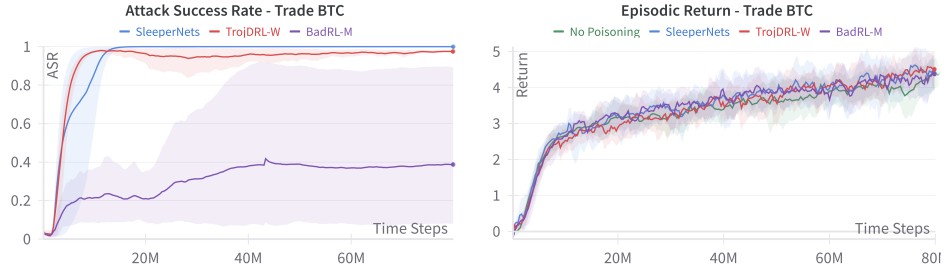

Figure 11: Best results for each attack on Trade BTC.

## 9.6 Poisoning Rate Annealing

In this section we explore the capabilities of SleeperNets to anneal its poisoning rate on many environments. In particular we implement poisoning rate annealing as follows – if the adversary's ASR is currently 100%, skip poisoning the current time step/trajectory. This method is applied to all three attack including TrojDRL and BadRL. SleeperNets is able to utilize our poisoning rate annealing the most, however, as it regularly achieves 100% ASR early in training. In Figure 12, Figure 13, and Figure 14 we present plots of the empirical poisoning rate of each attack while targeting each respective environment. We can see that the SleeperNets attack is able to decrease its poisoning rate to be significantly lower than all other attacks on every environment except for Highway Merge. Of particular note we achieve a 0.001% poisoning rate on Breakout and a 0.0006% poisoning rate on Trade BTC.

We can see that the poisoning rate of BadRL-M also often falls below the given poisoning budget, and falls very low in the case of Highway Merge. This is a feature of the attack as directly stated in [4]. One of the attack's main goals is to utilize a "sparse" poisoning strategy which minimizes poisoning rate while maximizing attack success and stealth. In many cases this can work, however in the case of environments like Highway Merge this works against the attack. In particular, the attack only poisons in states it sees as having a "high attack value" meaning the damage caused by taking $a^+$ instead of the optimal action is maximized. In environments like Highway Merge this set of states is very small and thus the attack only poisons occasionally, rather than regularly and randomly like TrojDRL and SleeperNets. We can see something similar in the case of Q*bert, however the attack eventually corrects for the low poisoning rate and ends up with a value near the budget of 0.03%.

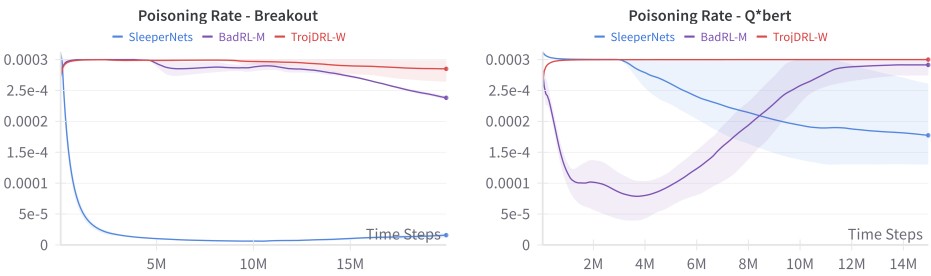

Figure 12: Comparison of poisoning rate for each attack on Breakout and Q*bert

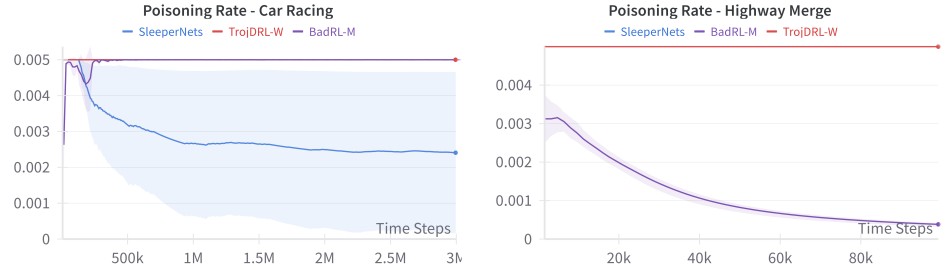

Figure 13: Comparison of poisoning rate for each attack on Car Racing and Highway Merge

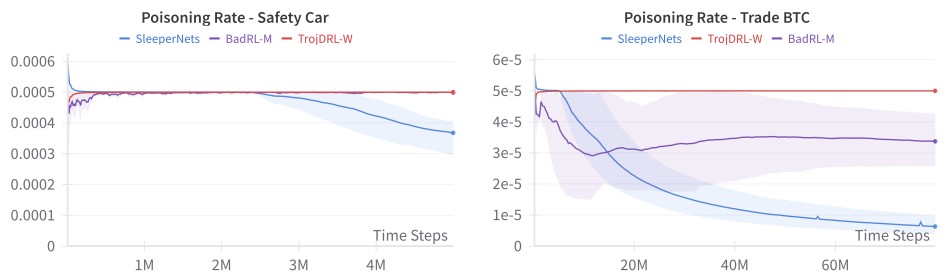

Figure 14: Comparison of poisoning rate for each attack on Safety Car and Trade BTC

## 9.7 Numerical Results from $c$ and $\beta$ Ablations

In this section we provide full numerical results from the ablations with respect to $c$ and $\beta$ presented in Section 6.3. All results are given with standard deviations in their neighboring column. The best result from each column is placed in bold.

| Ablation of Backdoor Attacks With Respect to $c$ | | | | | | | | | | |
|---|---|---|---|---|---|---|---|---|---|---|
| $c$ | 10 | | 20 | | 30 | | 40 | | 50 | |
| Metric | ASR | $\sigma$ | ASR | $\sigma$ | ASR | $\sigma$ | ASR | $\sigma$ | ASR | $\sigma$ |
| **SleeperNets** | **83.6%** | 9.4% | **98.9%** | 1.7% | **99.7%** | 0.4% | **100.0%** | 0.0% | **100.0%** | 0.0% |
| TrojDRL-W | 4.3% | 11.1% | 11.4% | 1.3% | 30.6% | 24.0% | 57.2% | 29.0% | 56.7% | 19.3% |
| BadRL-M | 0.1% | 0.0% | 0.2% | 0.1% | 0.1% | 0.0% | 0.2% | 0.1% | 0.1% | 0.0% |
| $c$ | 10 | | 20 | | 30 | | 40 | | 50 | |
| Metric | BRR | $\sigma$ | BRR | $\sigma$ | BRR | $\sigma$ | BRR | $\sigma$ | BRR | $\sigma$ |
| **SleeperNets** | 97.8% | 1.7% | 99.4% | 0.8% | 98.6% | 1.3% | 98.7% | 1.0% | 98.9% | 0.5% |
| TrojDRL-W | 98.2% | 1.0% | 94.7% | 3.2% | 95.1% | 2.9% | 90.4% | 7.9% | 90.8% | 3.8% |
| BadRL-M | **100%** | 0.0% | **100%** | 0.1% | **100%** | 0.0% | **100%** | 0.1% | **100%** | 0.0% |

Table 7: Numerical results from the ablation with respect to $c$. Each result is given with its respective standard deviation in the neighboring column. The best result in each column is placed in bold.

| Ablation of Backdoor Attacks With Respect to $\beta$ | | | | | | | | | | | |
|---|---|---|---|---|---|---|---|---|---|---|---|---|
| budget | 0.1% | | 0.25% | | 0.5% | | 1.% | | 2.5% | | 5% | |
| Metric | ASR | $\sigma$ | ASR | $\sigma$ | ASR | $\sigma$ | ASR | $\sigma$ | ASR | $\sigma$ | ASR | $\sigma$ |
| **SleeperNets** | 1.1% | 0.4% | **95.1%** | 1.8% | **100%** | 0.0% | **100%** | 0.0% | **100%** | 0.0% | **99.9%** | 0.2% |
| TrojDRL-W | 0.2% | 0.1% | 0.3% | 0.0% | 0.2% | 0.1% | 0.2% | 0.2% | 10.9% | 9.8% | 37.4% | 4.5% |
| BadRL-M | **4.7%** | 3.0% | 5.0% | 2.8% | 57.2% | 29.0% | 73.0% | 7.9% | 92.1% | 3.7% | 83.9% | 1.1% |
| budget | 0.1% | | 0.25% | | 0.5% | | 1% | | 2.5% | | 5% | |
| Metric | BRR | $\sigma$ | BRR | $\sigma$ | BRR | $\sigma$ | BRR | $\sigma$ | BRR | $\sigma$ | BRR | $\sigma$ |
| **SleeperNets** | 99.9% | 0.0% | 98.8% | 1.8% | 99.1% | 0.3% | 96.5% | 7.9% | 98.0% | 2.2% | 77.4% | 36.3% |
| TrojDRL-W | **100%** | 0.0% | **100%** | 0.2% | **100%** | 0.0% | **100%** | 0.0% | **99.4%** | 0.7% | **93.7%** | 6.6% |
| BadRL-M | 98.9% | 0.1% | 97.1% | 2.6% | 88.7% | 17.6% | 82.4% | 21.2% | 89.1% | 9.1% | 64.8% | 12.2% |

Table 8: Numerical results from the ablation with respect to $\beta$. Each result is given with its respective standard deviation in the neighboring column. The best result in each column is placed in bold.

## 9.8 Computational Resources Used

All experiments run in this paper are relatively low cost in terms of computational complexity – many can finish within a few hours running on CPU alone. For this paper all experiments were split between multiple machines including a desktop, laptop, and CPU server. Their specs are summarized in Table 9.

| Machines Used in Experimental Results | | | |
|---|---|---|---|
| Machine | CPU | GPU | RAM |
| Laptop | i9-12900HX | RTX A2000 | 32GB |
| Desktop | Threadripper PRO 5955WX | RTX 4090 | 128GB |
| Server | Intel Xeon Silver 4114 | None | 128GB |

Table 9: Summary of machines used for the experiments in this paper.

## 9.9 Related Work

In this section we discuss two alternate threat vectors against DRL algorithms – policy replacement attacks and evasion attacks. Note that "policy replacement attacks" is a term we are adopting for clarity, in the existing literature they are referred to generically as "poisoning attacks" [30]. We believe the term "policy replacement" is appropriate as will become evident. In policy replacement attacks the adversary's objective is to poison the agent's training algorithm such that they learn a specific adversarial policy $\pi^+ : S \to A$. Note that these attacks, unlike backdoor attacks, do not alter the agent's state observations at training or deployment time as they have no goals of "triggering" a target action. They instead only alter the agent's reward and or actions during each episode. Policy replacement attacks were extensively and formally studied in [30] and [29].

Evasion attacks, on the other hand, use either black-box [22] [2], or white-box [21] [3] access to the fully trained agent's underlying function approximator to craft subtle perturbations to the agent's state observations with the goal of inducing undesirable actions or behavior. These attacks have proven highly effective in DRL domains with multiple different observation types, such as proprioceptive [7] and image-based observations [19]. These attacks have a few main drawbacks. First, they incur high levels of computational cost at inference time - the adversary must solve complex, multi-step optimization problems in order to craft their adversarial examples [2] [21]. Second, applying the same adversarial noise to a different observation often does not achieve the same attack success, requiring a full re-computation on a per-observation basis. Lastly, many adversarial examples are highly complex and precise, and thus require direct access to the agent's sensory equipment in order to effectively implement [28].

