# OpenReview forum: "SleeperNets: Universal Backdoor Poisoning Attacks Against  Reinforcement Learning Agents"
_NeurIPS.cc/2024/Conference — NeurIPS 2024 poster_

### Official Review · Reviewer_Es9J · 2024-07-09

**Soundness:** 3
**Presentation:** 3
**Contribution:** 3
**Rating:** 7
**Confidence:** 3

**Summary:**

This paper proposes a new backdoor attack against Reinforcement Learning, termed SleeperNets. SleeperNets adopted dynamic reward poisoning to overcome the insufficiency of static reward poisoning proposed in previous works. The author provided a theoretical analysis of the advantages of dynamic adversarial poisoning and also conducted comprehensive evaluations to demonstrate the effectiveness of SleeperNets over previous backdoor attacks against DRL.

**Strengths:**

* This paper shows the drawback of static reward poisoning adopted in the previous DRL backdoor attacks, which motivates dynamic reward poisoning.
* The authors provide a theoretical analysis of the returned reward given the design of the dynamic reward, convincingly show that dynamic reward poisoning overcomes the drawbacks of static design.
* Comprehensive evaluations including detailed ablation studies are conducted.

**Weaknesses:**

NA

**Questions:**

NA

---

> ### Author Rebuttal · Authors · 2024-08-06
>
> **Overview:** We thank the reviewer for their strongly positive assessment of our paper. We appreciate them for acknowledging the many theoretical and technical contributions of our paper such as showing “the drawback of static reward poisoning”, providing “a theoretical analysis of the advantages of dynamic adversarial poisoning”, and in conducting “a comprehensive evaluation to demonstrate the effectiveness of SleeperNets”.
>
> Although the reviewer provided no weaknesses or questions for our paper, we encourage them to bring up any further questions they may have during the author discussion period.

---

### Official Review · Reviewer_YkW9 · 2024-07-12

**Soundness:** 3
**Presentation:** 3
**Contribution:** 3
**Rating:** 7
**Confidence:** 3

**Summary:**

The SleepNets paper considers a new ("outer loop") threat model, more powerful than those typically considered in adversarial RL settings.
The authors consider a stealthy attacker, aiming to both be successful (essentially tricking the learner into believing the underlying MDP is instead one of the attacker's choosing) and remain hidden (keeping the values of the original and corrupted policies similar).
They provide theoretical results on the limits of the traditional, weaker threat model, and introduce a "Dynamic Reward Poisoning Attack Formulation".
This yields their new attack "SleeperNets", which they empirically evaluate.

**Strengths:**

1. The paper has an explicitly stated threat model -- a welcome sight in this area.
2. The paper provides a theoretical investigation of prior threat models, with an ultimately simple example demonstrating an impossibility result.
3. While it is too far out of my area for me to be sure of the coverage of related work, the paper does seem to well-situate its contributions in the broader body of literature.
4. There is a broad but not overly cumbersome set of empirical analyses.
5. The paper is well written and easy to follow.

**Weaknesses:**

The main weakness I see is the applicability of the threat model.
As the others state, the adversary is assumed to infiltrate the computer on which the agent is training.
It's not clear to me what scenarios would exist where an attacker has that much access and can't perform a far deadlier attack (simply manipulating values directly).
The paper would be improved if the authors gave examples of real settings where an attacker could act in this outer-loop way without having direct software access.
I do believe such examples exist, they just need to be articulated.
That is, described in detail with specifications about how Algorithm 1 could still be executed (and the assumptions about e.g., \beta hold).

Some rough ideas for such settings:
1. The RL agent is acting on financial markets and the attacker is able to manipulate the reward signal by directly purchasing shares at an inflated cost from the agent.
2. The RL agent is acting in a physical environment and the attacker is able to manipulate that same environment (I'm picturing how humans train drug sniffing dogs by hiding toys).
3. The RL agent is flying, and the attacker has limited access to some of its instrumentation (e.g., can spoof its GPS location or jam certain signals).

**Questions:**

Same as the weakness described above -- in what real world settings is the attacker powerful enough to perform the sleepernets attack, but not so powerful as to directly manipulate memory values on the training machine?

**Limitations:**

The primary limitation is the weakness described above.
The other limitations are described in Section 7, and I agree with the authors that they are interesting areas for future work.
This paper stands without a deeper investigation of them.

---

> ### Author Rebuttal · Authors · 2024-08-06
>
> **Overview:** We thank the reviewer for their positive assessment of our paper and for their insightful questions. The reviewer’s main concern was the feasibility of the outer loop threat model. This is something we’ve taken much time to consider over the course of writing our paper, thus we will give the reviewer a thorough response.
>
> Upon reading our in-depth responses to all their questions, we encourage the reviewer to consider increasing their rating of our paper, or to engage in further discussion with us over the next week.
>
> **Comment 1:** “It's not clear to me what scenarios would exist where an attacker has that much access and can't perform a far deadlier attack (simply manipulating values directly).”
>
> **Response:** Here we focus particularly on the hypothetical, “far deadlier attack”’. First, we would like to ask the reviewer how they envision this attack? We lack a direct answer at this time, but encourage the reviewer to discuss this point further with us as it will be enlightening for all parties. For this response we will be assuming that the adversary has full control over the agent’s training machine and can arbitrarily manipulate RAM values. We will then push back on the notion that a “far deadlier attack” is either possible or sensible.
>
> When studying poisoning attacks we assume the adversary wishes to exploit the agent at test time, thus the agent must perform well enough in the base MDP to reach a deployment phase. In other words, the adversary must maintain “attack stealth” (section 3). This puts heavy restrictions on the adversary as they cannot simply replace the agent’s policy with one of their own design without training a model themselves. They must have full access to the victim’s MDP as well as the necessary compute resources to train the agent (which is hard and expensive to obtain). Accessing the MDP may be easy if it is simulated, but would be extremely difficult and costly in the case of real-world training environments, such as self-driving or robotic applications.
>
> Therefore, the most sensible option for the adversary is to implement a versatile attack like SleeperNets. In our work we show that extra information and domain specific engineering is largely unnecessary for a successful attack. So long as the adversary knows the agent’s observation space and can devise a trigger pattern they will be able to perform a successful and stealthy attack. Thus, we assert that the SleeperNet attack is a versatile and reliable approach for attackers of different capabilities.
>
> **Comment 2:** “The paper would be improved if the authors gave examples of real settings where an attacker could act in this outer-loop way without having direct software access…”
>
> **Response:** We thank the reviewer for this question since it is very fundamental to our paper. We additionally thank them for the inspirational examples they include in their comment. This question covers a very important topic, so we intend to answer it thoroughly and from multiple angles.
>
> **Viability of Direct Software Access:** We would like to first push back against the assertion that direct software access is unreasonable. There are countless, well documented cases of advanced persistent threats (APTs) achieving such levels of access in real-world settings. In fact, the most recent Verizon Data Breach Investigation (DBIR) report mentioned that in 2023 there were more than 5000 breaches with system intrusions, and these numbers are growing at a fast rate year-to-year. There are also infamous cases such as the RSA breach which prove that critical assets, such as model training servers, are not safe from adversarial attacks.
>
> **Applications of the Outer Loop Attack:** One nice feature of the outer loop threat model, in contrast to the inner loop, is that it allows for more direct translations between offline and online reinforcement learning. When attacking offline RL we can directly apply Algorithm 1 to the offline dataset intended to be used for training. The only difference is that, on line 2, we would sample H from the fixed dataset rather than from the MDP.
>
> Similarly, in many domains agents are trained with pseudo-MDPs created from existing real-world data. For instance, companies designing stock trading agents utilize real-world market data to train their models. This database, just like any other, is subject to direct manipulation as the result of a breach - which we know, from the aforementioned DIBR report, is unfortunately common.
>
> **Malicious Trainers:** Often in adversarial machine learning we assume the innocence of the training entity and assert the adversary must be external, however this isn’t always the case. We are currently working with collaborators on one such scenario. Imagine a company who designs and optimizes 5G connectivity controllers using DRL. Internet Service Providers will purchase products with the highest customer satisfaction, generally measured by how fairly they distribute internet bandwidth. However, the designer of the controllers may want to give preferential treatment to particular services (streaming platforms, etc.). How can they convince the ISP that their controller is fair while also allowing for this preferential treatment?
>
> A powerful solution to this problem is for the trainer to perform a backdoor attack against their own model - a setting they have complete control over. Through this method the trainer can guarantee state of the art performance - leading to the purchase of their controllers - while also allowing for exploitation of the backdoor after deployment. This solution requires no special tuning of the training algorithm, the model weights, or the MDP - it works directly “out of the box”. This necessitates further study into the auditing of DRL agents and the test-time detection of backdoor attacks, which we believe are both exciting areas of future research.

---

> > ### Comment · Reviewer_YkW9 · 2024-08-08
> > **Response to rebuttal**
> >
> > I thank the authors for their additional comments. These have helped clarify the paper and its context for me. I recommend that more of the discussion the authors laid out be added to the manuscript as space constraints allow, and I have increased my rating of the paper.

---

> > > ### Author Response · Authors · 2024-08-08
> > > **Thank you for the response**
> > >
> > > We are happy to hear that our response helped clarify the paper for you, and we're very grateful for your decision to increase your rating of our paper. We agree that including this additional context will be important for readers to understand our work, so we will be sure to make the proper additions to the threat model section of our paper.

---

### Official Review · Reviewer_VsP1 · 2024-07-13

**Soundness:** 3
**Presentation:** 3
**Contribution:** 3
**Rating:** 6
**Confidence:** 4

**Summary:**

The authors introduce a novel framework for backdoor poisoning RL agents, SleeperNets. SleeperNets assumes that adversaries can inject adversarial perturbations into the agent's observations during policy training within some total budget. Unlike in prior frameworks, the adversary implements its attacks post-hoc on full episode rollouts. The authors show that their attack manages to be successful, while retaining the performance of the optimal unpoisoned setting. The authors implement their novel framework on four different environments and show that it works favourably.

**Strengths:**

The paper has a number of strengths. First of all, I believe that the threat model innovations are sensible; it seems natural that the adversary could manipulate whole episodes and not just single steps. Equivalently, interpreting stealth as retaining policy performance seems sensible.

The authors' insight that "dynamic" attacks can attain both success and performance while "static" attacks cannot is insightful. The empirical results seem to support the author's claims.

**Weaknesses:**

* test-time defenders can still perform anomaly detection based on observations, actions or state transitions; I do think the authors should perform empirical investigations using out-of-distribution anomaly detection methods [2] to infer the information-theoretic detectability of their methods.

* The idea of condition of increasing the adversaries' attack context beyond single steps is not entirely novel within the adversarial attack literature, see e.g. [1] who devise adversarial attacks that condition on the entire action-observation history.

### minor weaknesses
line 264: "environment environment"

[1] Franzmeyer et al., Illusory Attacks, https://openreview.net/forum?id=F5dhGCdyYh, ICLR 2024

[2] Nasvytis et al., DEXTER, https://arxiv.org/abs/2404.07099, AAMAS 2024

**Questions:**

* you are not referencing adversarial cheap talk [3] - can you compare and contrast their setting against yours?

* you are mentioning a adversarial perturbation budget - where does this budget come from, and why would a budget be justified in reality rather than say a constraint based on information-theoretic detectability as in [1]?

[3] Lu et al., Adversarial Cheap Talk, ICML 2023

**Limitations:**

The authors do not further investigate defenses against their novel attack, although they state that securing the training environment as well as developing test-time anomaly detectors would be suitable avenues.

## Update in Response to the Rebuttal

The reviewer have successfully addressed my concerns, I therefore now recommend the paper for acceptance.

---

> ### Author Rebuttal · Authors · 2024-08-06
>
> **Overview:** We thank the reviewer for their positive assessment of our work in noting the sensibility of our threat model the insight brought by our theoretical analysis. The reviewer also brought with them much insight in the form of constructive questions and citations. We thank the reviewer for these comments and aim to respond to all of them  thoroughly. Given our in-depth responses, we encourage the reviewer to consider increasing their rating of our paper, or to engage in further discussion with us over the next week.
>
> **Comment 1:** “The authors implement their novel framework on four different environments and show that it works favourably.”
>
> **Response:** We thank the reviewer for their positive comments about our framework’s novelty and the favorability of our results. We would like to make a minor correction - we performed experiments over **6 different environments** which we then classified into 4 generic categories.
>
> **Comment 2:** “The idea of increasing the adversaries' attack context beyond single steps is not entirely novel within the adversarial attack literature, see e.g. [1]...”
>
> **Response:** We thank the reviewer for bringing [1] to our attention as it takes an approach to avoiding test-time detection which is very unique. We will certainly include it in an extended related work if our paper is accepted. That being said, it considers both a threat model and attack type - test-time evasion attacks - that are completely different from ours. While there are some similarities in giving the adversary access to more temporal information, our contribution of the outer-loop threat model is sufficiently novel and important to the study of backdoor attacks against DRL.
>
> Our paper is the first to introduce and formalize the outer-loop threat model for poisoning attack in RL - even when considering both backdoor and policy replacement attacks. We show that the outer loop threat model is not only viable and realistic, but allows for more powerful and dangerous attacks compared to the inner loop threat model. This necessitates further research into not only the capabilities and applications of the outer loop threat model, but also prevention against it.
>
> **Comment 3:** “you are not referencing adversarial cheap talk [3] - can you compare and contrast their setting against yours?”
>
> **Response:** We thank the reviewer for bringing up this work and giving us an opportunity for further reflection. We will definitely include it in an extended related work upon acceptance of our paper as an alternative class of poisoning attacks.
>
> In terms of threat model the papers are quite different. In [3] it is assumed the adversary can use an open “cheap talk” channel to send the agent “messages” **at each time step** during training and testing. The goal of each message is to influence the agent’s training behavior and potentially control them at test time. These messages are unique per state, requiring the adversary to learn a message function.
>
> In SleeperNets we instead allow the trigger to exist within the agent’s latent observation space - making no critical assumptions about the MDP. We additionally achieve state of the art results while poisoning less than 0.5% of the agent’s observations, while in [3] they use an effective poisoning rate of 100%. Lastly, in SleeperNets the adversary is allowed to manipulate the agent’s reward signal while [3] only perturbs the agent’s observations.
>
> We look forward to any future works attempting to combine the two attack methodologies or settings.
>
> **Comment 4:** “you are mentioning an adversarial perturbation budget - where does this budget come from?”
>
> **Response:** The concept of a poisoning budget (alternatively called the poisoning rate) is standard throughout the poisoning attack literature in both RL [4] and supervised learning [5]. There are two key reasons for its usage: we want to minimize the likelihood that the perturbations are detected at training time - which will increase with our poisoning budget; and we want the agent to still learn the benign MDP - which empirically becomes more difficult as one increases the poisoning budget (see our ablations in section 6.3).
>
> **Multiple Comments: Test Time Anomaly Detection via Information Theory**
> e.g. “...I do think the authors should perform empirical investigations using out-of-distribution anomaly detection methods [2] to infer the information-theoretic detectability of their methods.”
>
> **Response:** We thank the reviewer for bringing up [2] as it is an insightful work for the detection of test-time anomalies. We will certainly include a citation of this paper in an extended related work if our paper is accepted. We agree with the reviewer that the test-time detection of backdoor attacks is a critical and open problem in this field, but we believe it is orthogonal to the objectives of our paper.
>
> In our paper we answer the fundamental question “What poisoning methods are necessary for successful RL backdoor attacks?”. We not only prove the insufficiency of static reward poisoning, but we also develop a novel, dynamic reward poisoning strategy and rigorously prove its sufficiency in achieving both attack success and stealth - becoming the first backdoor attack paper in RL to produce such concrete results. Through this we propose a versatile attack framework which makes no critical assumptions about the attacker’s test time objectives. This allows future poisoning attack literature in RL to build upon our theoretically rigorous foundations and technically rich contributions when they aim to solve additional adversarial objectives, such as minimizing the detectability of test-time anomalies in the agent’s observations or actions.
>
> [4] Kiourti et. al., TrojDRL: Trojan Attacks on Deep Reinforcement Learning Agents. ACM/IEEE Design Automation Conference (DAC), 2020
>
> [5] Shafahi et. al., Poison Frogs! Targeted Clean-Label Poisoning Attacks on Neural Networks. NIPS 2018

---

> > ### Comment · Reviewer_VsP1 · 2024-08-12
> > **Thanks for clarifying my concerns.**
> >
> > I thank the reviewers for clarifying my concerns; I have decided to increase my score in response.

---

### Author Rebuttal · Authors · 2024-08-06

We would like to first extend our thanks to the reviewers for their time in reading our paper, evaluating its merits, and highlighting its novel contributions. We hope that our extensive responses have sufficiently answered all the reviewers’ questions, but openly invite any further questions or comments during the author discussion period.

We are further grateful for the reviewers’ enlightening comments and feedback. Reviewer VsP1 was primarily concerned with the test time detectability of our attack against information theoretic defenders, citing multiple related works. In our response we highlight the versatility of our attack framework and its applicability to these additional, adversarial objectives. We further note the relevance of their cited papers and aim to include them in an extended related work.

Reviewer YkW9 asked questions about the viability of our threat model in the real world, additionally including some insightful attack scenarios. We answer these questions thoroughly and from multiple angles. We first establish the utility of our attack for both strong and weak adversaries, highlighting the difficulty and extra cost of implementing a “far deadlier attack” without being detected. We then provide multiple use cases of the outer loop attack and motivate its positioning within both the literature and real world settings.

Reviewer Es9J was highly positive of our work, noting their confidence in both our theoretical contributions and our empirical evaluation. We greatly appreciate this praise and openly invite the reviewer to engage with us in further discussion during the author rebuttal period if they have any additional questions or comments.

Finally, when writing this paper we aimed to take a theoretically rigorous approach towards filling crucial gaps in our understanding of backdoor attacks in reinforcement learning. Through this exploration we not only prove the insufficiency of prior approaches, but make key contributions in developing the first backdoor attack framework to provably maximize both attack success and attack stealth. We believe these theoretical developments, in addition to our novel SleeperNets attack and outer loop threat model, will be a crucial foundation for future works studying both backdoor attacks and defenses in reinforcement learning.

---

### Decision · Program_Chairs · 2024-09-25

**Decision:**

Accept (poster)

**Comment:**

The paper proposes a novel universal backdoor poisoning attack.  The reviewers appreciated the novelty and technical contribution which combines theoretical insights with a practical algorithmic approach that is shown to be highly effective in experiments.